# Genome-wide functional screening of drug-resistance genes in *Plasmodium falciparum*

Shiroh Iwanaga [1,2] ✉, Rie Kubota[3], Tsubasa Nishi[4], Sumalee Kamchonwongpaisan[5], Somdet Srichairatanakool[6], Naoaki Shinzawa[3], Din Syafruddin[7], Masao Yuda[4] & Chairat Uthaipibull[5,8]

The global spread of drug resistance is a major obstacle to the treatment of *Plasmodium falciparum* malaria. The identification of drug-resistance genes is an essential step toward solving the problem of drug resistance. Here, we report functional screening as a new approach with which to identify drug-resistance genes in *P. falciparum*. Specifically, a high-coverage genomic library of a drug-resistant strain is directly generated in a drug-sensitive strain, and the resistance gene is then identified from this library using drug screening. In a pilot experiment using the strain Dd2, the known chloroquine-resistant gene *pfcrt* is identified using the developed approach, which proves our experimental concept. Furthermore, we identify multidrug-resistant transporter 7 (*pfmdr7*) as a novel candidate for a mefloquine-resistance gene from a field-isolated parasite; we suggest that its upregulation possibly confers the mefloquine resistance. These results show the usefulness of functional screening as means by which to identify drug-resistance genes.

*Plasmodium falciparum* is the most virulent human malaria parasite; indeed, it is responsible for the majority of malaria deaths, which are estimated to reach more than half a million annually[1,2]. Patients are currently treated using artemisinin combination therapy (ACT), in which artemisinin is used as a first-line drug along with partner drugs such as mefloquine, piperaquine, and lumefantrine[3]. ACT has greatly contributed to the decrease in malaria deaths over the past two decades[1,2]; however, since 2009, treatment failure caused by resistance to the drugs used during ACT has been reported in endemic areas including South-East Asia and Africa[4,5]. Therefore, there is concern that the therapeutic effects of ACT may be decreasing[6].

Identification of drug-resistance genes contributes to not only the elucidation of molecular mechanisms of resistance but also the development of molecular markers for surveillance; thus, it offers one potential solution to the problem of drug resistance[7]. When studying *P. falciparum*, the current approach used to identify drug-

resistance genes is genome-wide sequencing analysis of laboratory-produced drug-resistant parasites; specifically, drug-sensitive parasites are exposed to long-term drug pressure, and the parasite that newly acquires resistance by natural mutations is selected from them, the mutations are eventually identified using single nucleotide polymorphism analysis via next-generation sequencing (NGS)[8]. This approach was utilized to identify the artemisinin-resistance gene *kelch13* (PF3D7_1343700), and the mutations found in this gene have been used for the surveillance of artemisinin-resistance[9]. However, it generally takes months to several years to obtain drug-resistant parasites using drug exposure techniques[10–16], which is one of the reasons why the identification of drug-resistance genes requires a long time. Therefore, an alternative approach that can be used to directly identify drug-resistance genes in parasites isolated from patients has been required to improve the speed and convenience of the process.

[1]Department of Molecular Protozoology, Research Institute for Microbial Diseases, Osaka University, Osaka, Japan. [2]Center for Infectious Disease Education and Research, Osaka University, Osaka, Japan. [3]Department of Parasitology and Tropical Medicine, Graduate School of Medical and Dental Science, Tokyo Medical and Dental University, Tokyo, Japan. [4]Department of Medical Zoology, Faculty of Medicine, Mie University, Mie, Japan. [5]National Center for Genetic Engineering and Biotechnology, National Science and Technology Development Agency, Pathum Thani, Thailand. [6]Department of Biochemistry, Faculty of Medicine, Chiang Mai University, Chiang Mai, Thailand. [7]Malaria and Vector Resistance Laboratory, Eijkman Institute for Molecular Biology, Jakarta, Indonesia. [8]Thailand Center of Excellence for Life Science, Bangkok, Thailand. ✉e-mail: iwanaga@biken.osaka-u.ac.jp

Functional screening is an approach used widely to identify target genes, which include not only drug-resistant genes but also the genes that are essential to various biological processes such as nuclear division, cell division, and the functioning of metabolic pathways. In a previous study, we demonstrated that a drug-resistance gene could be identified using functional screening in *Plasmodium berghei*, which is the rodent model of human malaria[17]. Briefly, we generated high-coverage genomic libraries of a drug-resistant parasite strain using a *Plasmodium* artificial chromosome (PAC)[18] in drug-sensitive parasites, and we then selected parasites that newly acquired resistance from the libraries using a drug-screening approach. Subsequently, we recovered the PAC with the inserted DNA fragments from the selected parasites, and we eventually identified drug-resistance genes by analyzing the inserted DNA fragments. The PAC can be used to introduce large DNA fragments, exceeding 10 kb in size, directly into parasites; this facilitates the construction of genomic libraries that represent the entire coding region of all parasite genes[17].

In this work, we develop a similar approach for identifying drug-resistant genes using functional screening of *P. falciparum*. We demonstrate that a genomic library could be generated using the centromere plasmid, which is expected to introduce a large DNA fragment similar to the PAC. In addition, we find that the chloroquine-resistant transporter (*pfcrt*; PF3D7_0709000) gene can be identified using drug screening of the genomic library from the laboratory strain. Furthermore, we isolate mefloquine-resistant *P. falciparum* from a patient living in a malaria-endemic area and identify a novel candidate for a mefloquine-resistant gene using our developed functional screening method.

## Results

### Genomic library construction in parasites

For the genome-wide screening of drug-resistance genes (Fig. 1), the genomic library must cover more than one genome equivalent. In addition, each insert DNA fragment must contain at least one gene; the length of fragments should be >5 kb because the average gene density of *P. falciparum* is 4338 bp[19]. First, we examined whether it was possible to generate a genomic library that would meet these criteria. In this study, we used the centromere plasmid, pFCENv1, which contained the centromere sequence from chromosome 5 of *P. falciparum*, as a vector[20]. The pFCENv1 would be expected to deliver large DNA fragments as well as PAC even without telomere sequence. In addition, this plasmid is maintained as single-copy DNA by the function of centromere. The partially digested genomic DNA of strain 3D7 was separated using agarose gel electrophoresis, and DNA fragments >10 kb in size were then purified. Five micrograms of those purified DNA fragments were subsequently ligated with the same amount of the pFCENv1 plasmid in vitro, and the resultant ligation reaction mixture was then directly used for transfecting highly synchronized schizonts, which contained 2-4% of full mature form, according to our recent published protocol[21]. The transgenic parasites were screened using pyrimethamine (25 ng/ml), as pFCENv1 had human dihydrofolate reductase as a drug-selectable marker. The parasites were always visible in culture during drug selection, implying successful transfection. (Supplementary Fig. 1a). Similar experiments were repeated twice more and transgenic parasites were obtained reproducibly (Supplementary Fig. 1a). To evaluate transfection efficiency, we estimated the number of independently transfected parasites according to the reported equation with modification using the percentages of parasitemia[22,23]: the number of the parasites at some time point can be calculated using the percentages of parasitemia and the multiplication rate, as described in Methods. The percentages of parasitemia in those three transfection experiments were microscopically estimated as 0.67, 0.53, and 0.40 at 14-days posttransfection (Supplementary Fig. 1a), and multiplication rate (3.7 per cell cycle) of transgenic parasite with pFCENv1 was used[20]. As a result of the calculation, 702, 562,

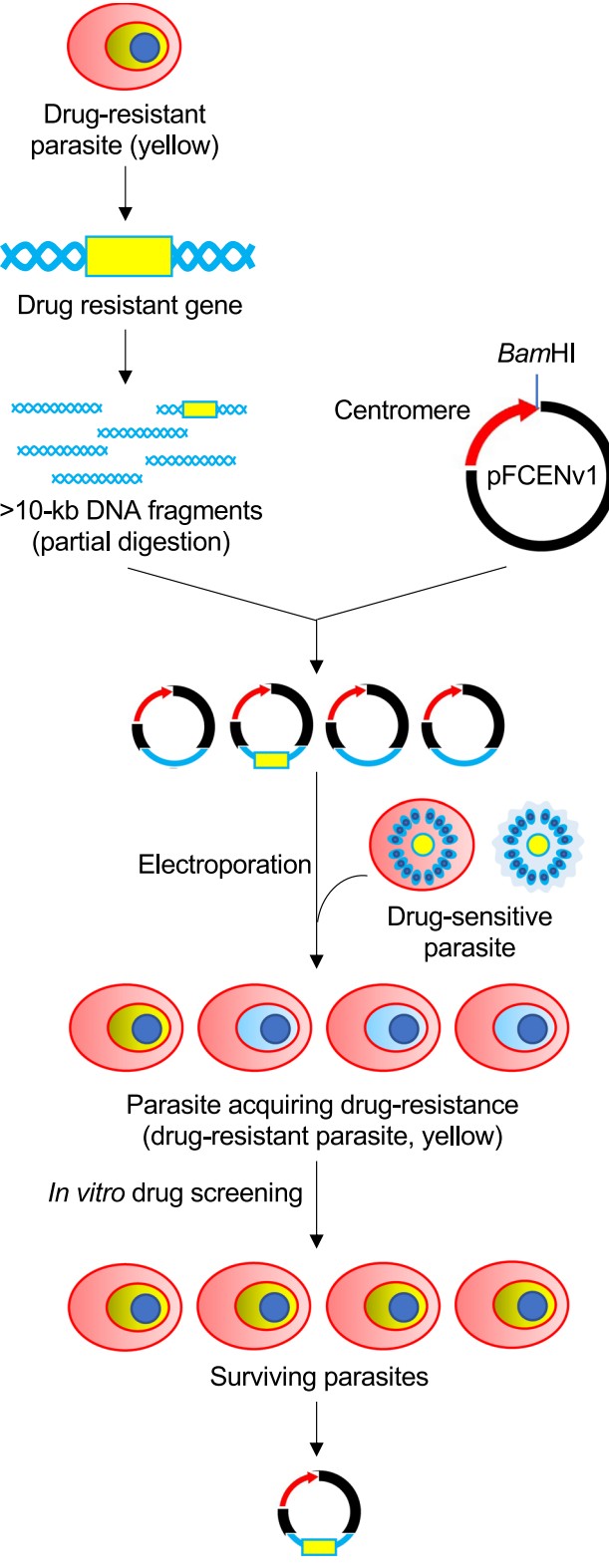

and 421 parasites were estimated to have been transfected independently in each experiment, and the average of those numbers were 561.7. We further examined the number of independently transfected parasites by another method: the 1/10 and $1/10^2$ of culture containing transfected parasites were dispensed into 48 wells immediately after electroporation; the numbers of the parasite-positive wells were

**Fig. 1 | Experimental scheme of the functional screening approach used to identify a drug-resistance gene from a single drug-resistant *P. falciparum* strain.** High-coverage genomic libraries were generated in drug-sensitive parasites (blue) from drug-resistant strains (yellow) using the centromere plasmid pFCENv1. The recipient parasite acquired drug resistance due to the introduction of the drug-resistance gene; thus, it survived during drug screening. The DNA fragment inserted in pFCENv1 was recovered from the surviving parasites, and its sequence was determined using genome-walking analysis.

counted after long-term cultivation; the number of independently transfected parasites were estimated by calculating using the those counted number of parasite-positive well (Supplementary Fig. 1b). We repeated this assay 10 times. When using the culture containing 1/10 of transfected parasites, they were detected in almost all wells, while when using $1/10^2$ of them, parasites were detected in 2–6 wells in each assay (Supplementary Fig. 1b). The average number of independently transfected parasites was roughly estimated as 413 and 400 in the assays using 1/10 and $1/10^2$ of them, respectively, which were roughly consistent with that estimated using the parasitemia and multiplication rate. Taking average numbers obtained from two different assays together, we estimated that at least 500 parasite clones were transfected independently using each 5-μg sample of partially digested DNA fragments and the centromere plasmid vector.

Subsequently, the average size of the insert DNA fragments was estimated using contour-clamped homogeneous electrical field (CHEF) analysis of the parasite clones obtained from the genomic library, followed by Southern blot hybridization. The signals in 10 parasite clones were detected at sizes in the range of 24.8–45.0-kb, including the size of the vector (8018 bp; Supplementary Fig. 1c). The average size of the insert DNA fragments was estimated to be 25.9 kb after subtraction of the vector size. Based on the gene density of *P. falciparum*[18] a single insert DNA fragment was estimated to encode six genes.

The genomic coverage of the generated library was calculated to be ~0.52 based on the number of independently transfected parasites (500 parasites per transfection), the average size of the DNA insert (25.9 kb), and the genome size of *P. falciparum* (25 Mb)[19]. This result showed that the genomic library, which represented approximately one genome equivalence, was generated from two repeated transfections with 5 μg of insert DNA fragments.

## Functional screening for chloroquine-resistance genes

Mutated *pfcrt* was identified as a chloroquine-resistance gene using forward genetic analysis of the progenies of *P. falciparum* strains Dd2 and HB3, which are chloroquine-resistant and chloroquine-sensitive, respectively[24]. We used the Dd2 to test whether a known drug-resistance gene, namely *pfcrt*, could be identified from the genomic library using functional screening. Three genomic libraries were generated independently, and the genomic coverage of each library was estimated to be ~2.6 genome equivalents. These genomic libraries were subsequently treated with 20-nM chloroquine for 4 days (Supplementary Fig. 2a); parasite survival was detected in two genomic libraries (dd2-1ib1 and 2) 3 days after chloroquine withdrawal. In contrast, all parasites in another library (dd2-1ib3) and the negative control parasite, in which only pFCENv1 was introduced, were killed by this treatment (Supplementary Fig. 2b). The surviving parasites in the two genomic libraries were treated again with 20-nM chloroquine for 4 days (Supplementary Fig. 2a, b), followed by further treatment with 20 nM chloroquine for 6 days (Fig. 2a). Parasite survival was detected in both genomic libraries, suggesting that some parasites had acquired chloroquine resistance and had been selected (Fig. 2a). A clonal parasite was established from each culture containing surviving parasites using limiting dilution, after which their 50% inhibitory concentration ($IC_{50}$) values were determined in independent quadruplicate assays.

The $IC_{50}$ values of clones obtained from dd2-1ib1 and 2 were estimated to be 159.1 ± 1.9 nM and 140.6 ± 2.2 nM, respectively. Although these values were 2.6-fold lower than that of the parental strain Dd2 (379.9 ± 4.4 nM), they were 13.9-fold higher than those of the 3D7 (11.4 ± 0.4 nM) and the negative control parasite (9.4 ± 0.6 nM), which confirmed that the strains had acquired resistance (Fig. 2b).

Subsequently, we recovered pFCENv1 from the two parasite clones and determined both sequence ends of the insert DNA fragments using genome-walking analysis (Supplementary Fig. 3a). A BLAST search using the determined sequences showed that the insert DNA fragments were detected at positions 399,830–411,518 and 391,912–416,877 on chromosome 7 (Fig. 2c and Supplementary Fig. 3b), indicating that the fragments overlapped each other. The overlapped region corresponded to the *pfcrt* gene and its regulatory elements such as the promoter and 3′ untranslated region (3′-UTR) (Fig. 2c). These results clearly showed that parasites that had newly acquired chloroquine resistance through the introduction of mutated *pfcrt* could be selected from genomic libraries using functional screening, and that *pfcrt* introduced into pFCENv1 could then be identified using sequencing analysis.

We further investigated the expression level of the *pfcrt* in the selected parasites. First, we performed the copy number (CN) analysis using quantitative polymerase chain reaction (qPCR) using the DNA recovered from the selected clonal parasites. The results indicated the presence of two *pfcrt* copies (Supplementary Fig. 4) in each of them. Each *pfcrt* copy originated from the 3D7 genome and the insert DNA in the pFCENv1. Next, the 3D7 and the Dd2 *pfcrt* promoter activities were compared via quantitative reverse transcription PCR (RT-qPCR) analysis using RNA isolated from each strain. This analysis showed that these activities were almost identical between the strains 3D7 and Dd2 (Supplementary Fig. 5). These results suggested that the expression of mutated PfCRT of the Dd2 was almost identical to that of wild-type PfCRT of the 3D7 in the selected parasites. Therefore, we considered that the mutated PfCRT was probably sufficient to confer resistance even at half abundance of the total PfCRT amount in the selected parasites.

In summary, based on the above results, we considered that drug-resistance genes could be explored in *P. falciparum* via functional screening in the genomic library generated using pFCENv1.

## Identification of a mefloquine resistance gene using functional screening

Mefloquine is widely used not only as a partner drug for artemisinin in ACT but also as a prophylactic drug. Resistance to mefloquine is known to be conferred by duplication of the multidrug resistance protein 1 (*pfmdr1*, PF3D7_0523000) gene in *P. falciparum*[25–28]. However, a copy number variation (CNV) study of parasites isolated in Thailand implied the involvement in mefloquine resistance of a gene other than *pfmdr1*: mefloquine resistance was identified in 16 of 85 field-isolated parasites harboring a single-copy of *pfmdr1*[29]. Thus, we attempted to identify a novel candidate for a mefloquine-resistance gene from a field-isolated parasite using functional screening. To this end, we isolated a mefloquine-resistant parasite, namely the strain MEF1, from a patient living in the Thai−Myanmar border area. It had an $IC_{50}$ of 33.0 ± 1.1 nM, which was approximately 2.4-fold higher than that of strain 3D7 (13.8 ± 0.6 nM). CNV analysis conducted using NGS showed that the MEF1 had single-copy of *pfmdr1* (Supplementary Data 1); thus, mefloquine resistance was not conferred in this strain by duplication of this gene.

Eight genomic libraries were generated using purified genomic DNA of the MEF1. The average genomic coverage of the libraries was estimated to be ~1.6; therefore, the sum of the genomic coverage was 12.8. These genomic libraries were treated twice with 15-nM mefloquine for 4 days (Supplementary Fig. 6a, b), and the surviving parasites were detected in three genomic libraries (mef-lib2, 3, and 6). They were

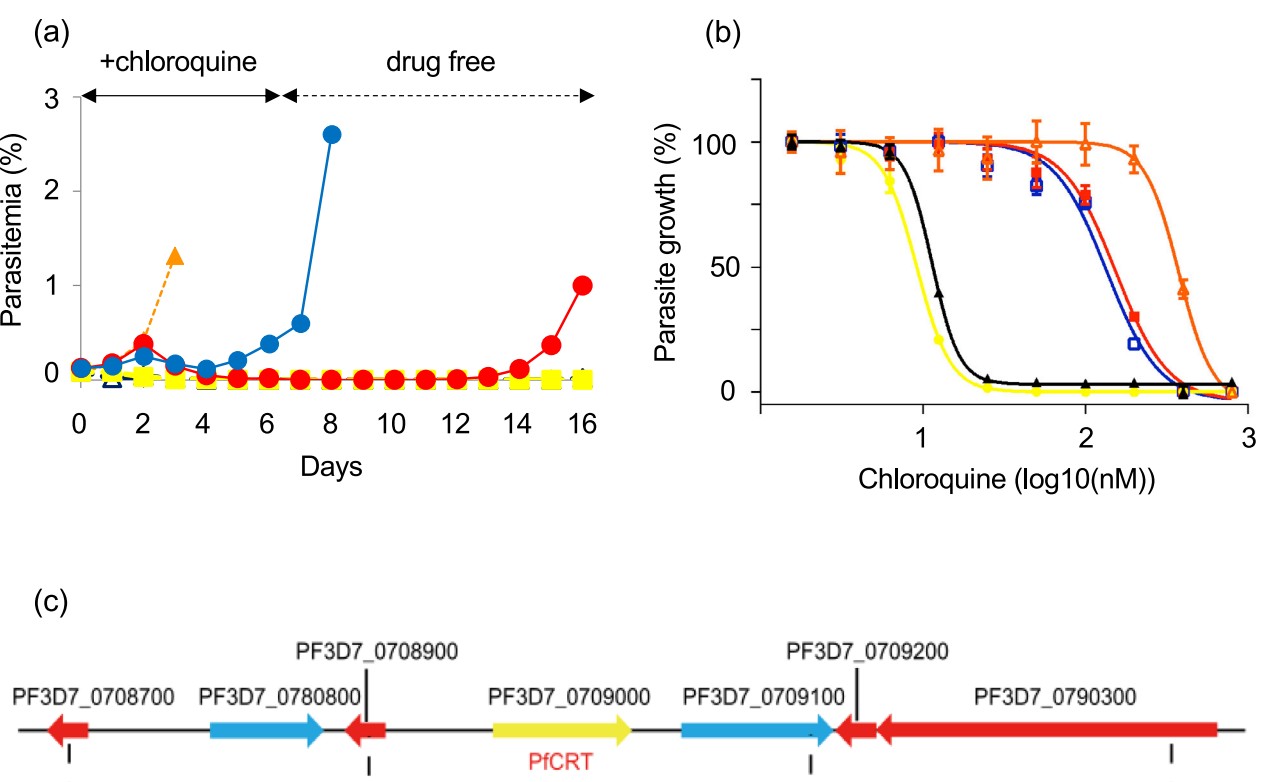

**Fig. 2 | Functional screening of a chloroquine-resistance gene from the genomic libraries of strain Dd2. a** Parasites [dd2-lib1 (red), 2 (blue), 3D7 (black), Dd2 (orange), and the negative control parasite (yellow)] were treated with 20-nM chloroquine for 6 days. The transgenic parasite, in which the pFCENv1 was introduced, was used as a negative control. **b** The $IC_{50}$ values of clonal parasites selected from dd2-lib1 and 2 were determined from The values for strains 3D7 and Dd2 and the negative control parasite was also determined. All assays were performed using $n = 4$ biological independent samples. Error bars are SEM. The color specification is the same as that used in **a. c** Genomic DNA fragments were obtained from surviving parasites in dd2-lib1 and 2. DNA fragments overlapped at the genomic region encoding the *pfcrt* gene. Source data are provided as a Source Data file.

further treated with 15-nM mefloquine for 6 days, and parasite survival was detected in two libraries (mef-lib3 and 6) (Fig. 3a). A clonal parasite was obtained from each culture containing surviving parasite, and the pFCENv1 with the insert DNA fragments was recovered from each parasite. Sequence analyses identified the insert DNA fragments at positions 436,938–465,288 and 438,880–455,299 on chromosome 12, indicating that they encoded the same genomic region (Fig. 3b and Supplementary Data 2). In addition, qPCR analysis of those selected parasite clones showed that the genes on the delivered insert DNA fragments were maintained as single-copy DNA (Supplementary Fig. 7). The $IC_{50}$ values of the two clones from mef-lib3 and mef-lib6 were determined as $21.1 \pm 1.6$ nM and $21.6 \pm 0.9$ nM, respectively, in independent quadruplicate assays (Fig. 3c). Although these $IC_{50}$ values were slightly lower than that of the parental MEF1, they were -1.6-fold greater than those of the 3D7 and the negative control parasite ($13.2 \pm 0.6$ nM); thus, mefloquine resistance was confirmed in these clones.

To examine whether this genomic region on chromosome 12 was reproducibly identifiable and to further delimit the candidate genomic

region, we generated eight additional genomic libraries of the MEF1 (Supplementary Fig. 8: mef-lib9–16) and screened the parasites for mefloquine resistance. Parasite survival was detected in four libraries (mef-lib10, 11, 12, and 13) in the second experiment (Supplementary Fig. 8), and each was used to establish a clonal parasite. Sequence analyses showed that two insert DNA fragments from the parasite clones derived from mef-lib10 and lib13 overlapped with the fragments identified in the first experiment at position 445,838–455,299 on chromosome 12 (Fig. 3b and Supplementary Data 2). These results strongly suggest that the mefloquine-resistance gene was encoded in this region. However, the recovered DNA fragments from another two parasite clones from mef-lib11 and mef-lib12 encoded different genomic regions at positions 655,248–667,533 on chromosome 14 and 507,747–528,992 on chromosome 11, respectively (Supplementary Fig. 8 and Supplementary Data 2). Although these fragments might be involved in mefloquine resistance, we did not investigate them further in the present study.

The identified genomic region (position 445,838–455,299) on chromosome 12 encoded three genes: a putative ATP synthase

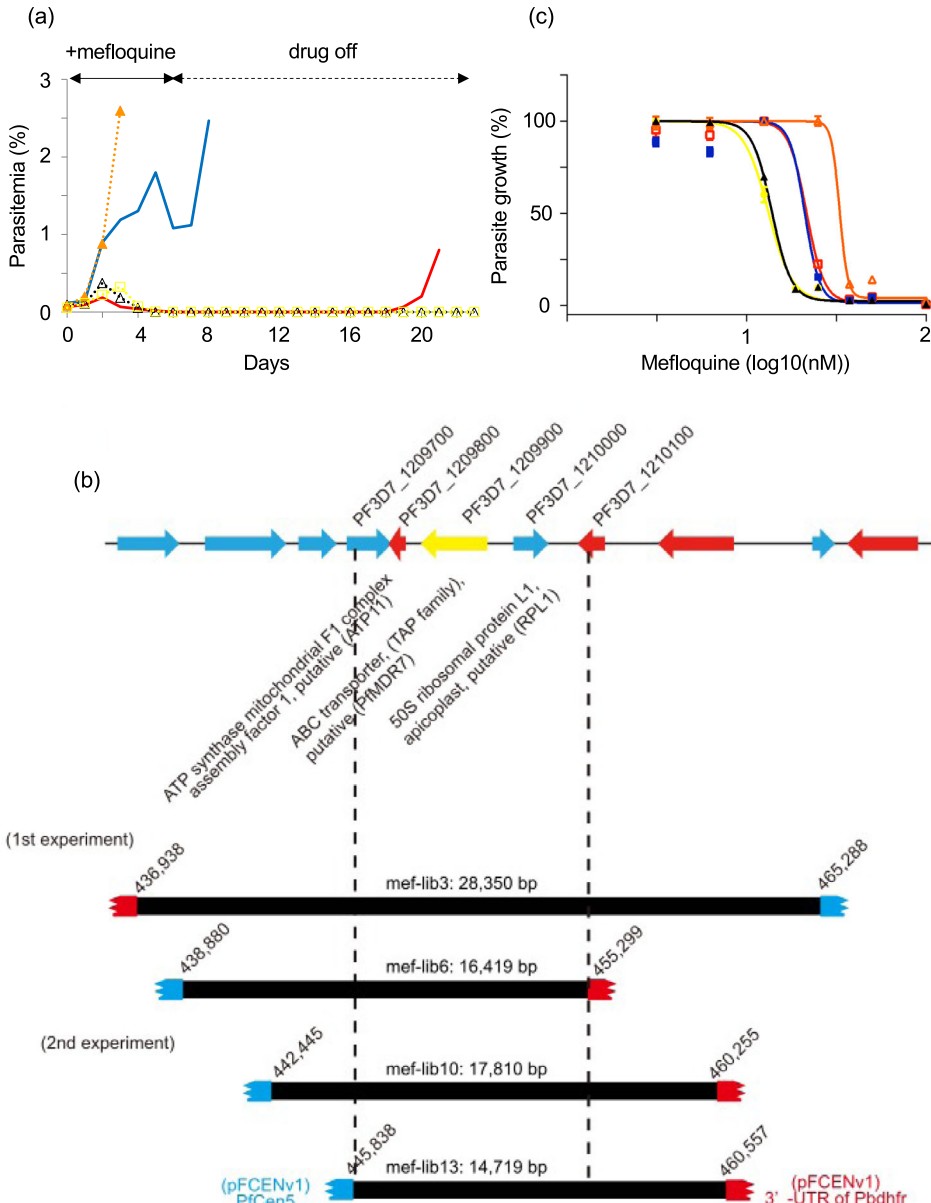

**Fig. 3 | Functional screening of a mefloquine-resistance gene from the genomic libraries of strain MEF1. a** Parasites in mef-lib3 (red), and 6 (blue), 3D7 (black), MEF1 (orange), and the negative control parasite (yellow) were treated for 6 days with 15-nM mefloquine. **b** Four genomic DNA fragments were recovered from the parasites, which were selected from mef-lib3, 6, 10, and 13. **c** The IC$_{50}$ values of clonal parasites selected from mef-lib3, mef-lib-6, 3D7, MEF1, and the negative control parasite were determined. All assays were performed using $n = 4$ biological independent samples. Error bars are SEM. The color specification is the same as that used in **a**. Source data are provided as a Source Data file.

mitochondrial F1 complex assembly factor 1 (PF3D7_1209800; *atp11*), a putative ABC transporter B (ABCB) family member 7 (PF3D7_1209900), and a putative apicoplast ribosomal protein L1 (PF3D7_1210000; *rpl1*) (http://plasmodb.org/plasmo/). To determine the genes responsible for mefloquine resistance, we assessed the mefloquine resistance of the transgenic parasites, in which each gene was introduced. We first amplified the coding sequence (CDS) of each gene along with their original promoter from MEF1, and cloned them into pFCENv1 using *E.coli*. The CDS of PF3D7_1209900 along with its promoter was successfully incorporated in the plasmid, which was then introduced into the 3D7 via electroporation. However, the amplified CDS and promoters of PF3D7_1209800 and PF3D7_1210000 could not be cloned in the pFCENv1 using *E.coli* because of their size and AT richness. Thus, we directly introduced the ligation reaction mixtures of those amplified fragments and pFCENv1 into the 3D7 and generated the clonal transgenic parasites with extra copies of each

gene. All transfection for generating each transgenic parasite was performed in duplicates. The results clearly showed that the putative ABC transporter B family member 7 conferred mefloquine resistance to the parasites (Fig. 4a and Supplementary Fig. 9), whereas the other two genes did not confer any resistance (Supplementary Fig. 10). These results suggested that this ABC transporter was potentially the mefloquine-resistance gene in the MEF1.

The identified ABC transporter is a member of the multidrug-resistant protein family in *P. falciparum*, and it has already been annotated as *pfmdr7*. Sequence analysis of the *pfmdr7* gene from the MEF1, compared with the amino acid sequence of the 3D7, revealed the insertion of three amino acids, Asn-Val-Arg, at position 27 and of five Asn residues at position 156. However, similar insertions were found in field-isolated mefloquine-sensitive parasites, suggesting that these might not be responsible for the resistance (Supplementary Data 3). To examine this, the CDS of *pfmdr7* of the 3D7 along with its promoter was

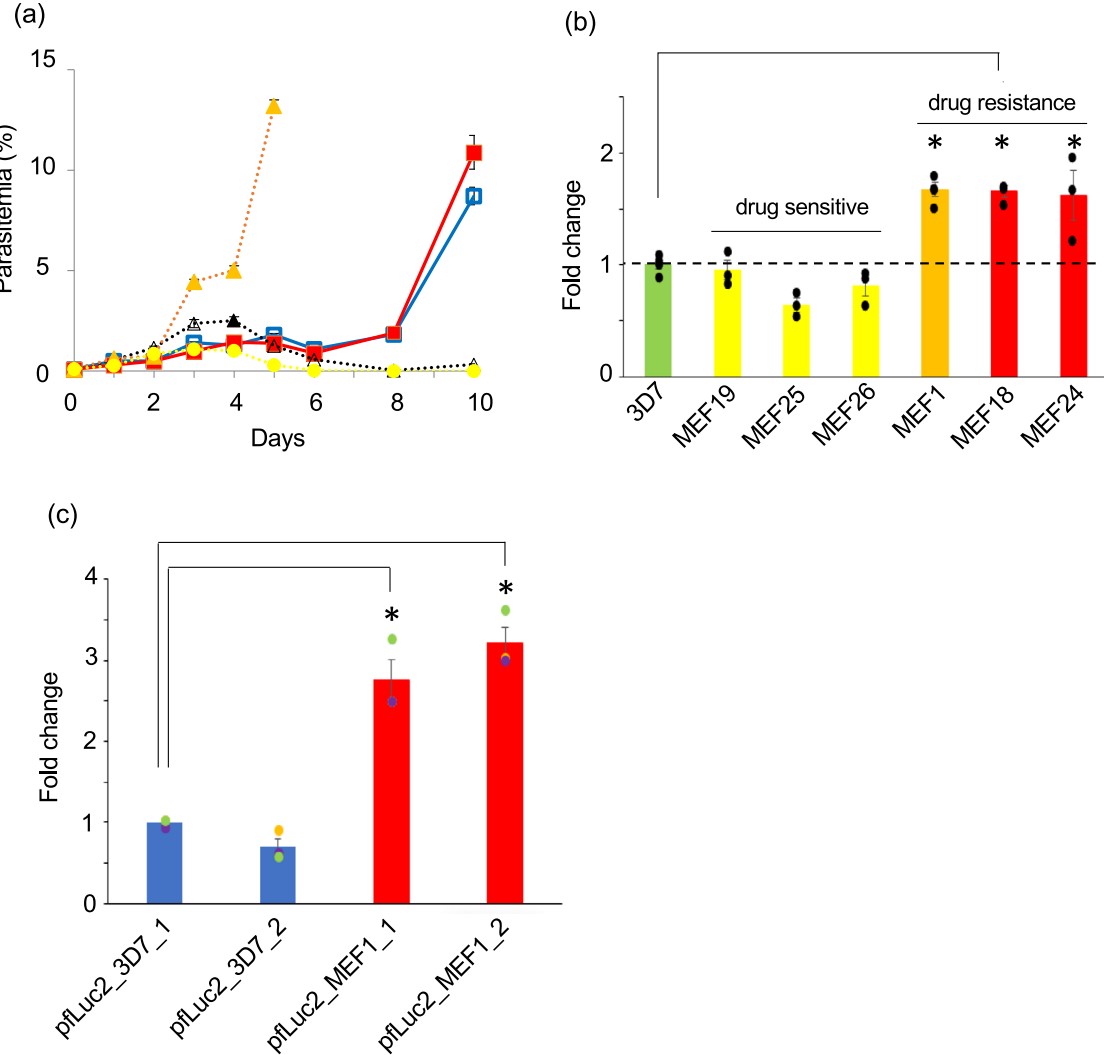

**Fig. 4 | Mefloquine resistance caused by PfMDR7. a** The mefloquine resistance of a transgenic parasite, into which *pfmdr7* of the MEF1 (red) and the 3D7 (blue) were introduced, were examined. In addition, the mefloquine resistance of the 3D7 (black), the MEF1 (orange), and the negative control parasite (yellow) into which only pFCENv1 was introduced were also examined. **b** The mRNA level of *pfmdr7* in the MEF1(orange) was analyzed by RT-qPCR. In addition, those of three mefloquine-sensitive (yellow) and two mefloquine-resistant parasites (red), which were collected from patients living in the Thai–Myanmar border area, were analyzed. Strains 3D7 (green) were used as a negative control. All assays were performed using *n* = 3 biological independent samples of each parasite strains. Error bars are SEM, and the measure of the center is the means value. The *P* values were calculated from statistical analysis with a two-sided Student's *t* test. Asterisks indicate *P* values <0.05 for comparisons between the 3D7 and the resistant parasites. **c** The promoter activities of the *pfmdr7* of the MEF1 (red) and the 3D7 (bule) were examined using the transgenic parasites, in which the plasmids having each promoter were introduced. Transfection was performed in duplicates for each plasmid. The relative changes of luminescence of transgenic parasites were estimated based on one transgenic parasite, in which the promoter of *pfmdr7* of the 3D7 was introduced. Assays were performed using *n* = 3 biological independent samples of transgenic parasites. Error bars are SEM, and the measure of the center is the means value. The *P* values were calculated from statistical analysis with a two-sided Student's *t* test. Asterisks indicate *P* values <0.005 for comparison between the samples. Source data are provided as a Source Data file.

introduced into the 3D7 using pFCENv1, and the mefloquine resistance of the resultant transgenic parasite was then assessed. This transgenic parasite exhibited mefloquine resistance, apparently, supporting that those insertions were not involved in the resistance (Fig. 4a). Furthermore, considering acquiring mefloquine resistance via introducing wild-type PfMDR7, this result suggests that PfMDR7 could inherently confer mefloquine resistance. Therefore, we considered that the transgenic parasites increased the ability of mefloquine resistance via the introduction of an extra *pfmdr7* copy, by which they acquired mefloquine resistance.

The CNV analysis of MEF1 showed that *pfmdr7* was a single-copy gene similar to that of 3D7 (Supplementary Data 1), indicating that MEF1 does not acquire mefloquine resistance by increasing the CN of *pfmdr7* gene. However, as transcriptional upregulation has a similar effect as CN increase, we examined the expression level of the MEF1

*pfmdr7* mRNA by RT-qPCR using four independent biological samples. The result revealed a clear increase (1.6-fold) in the *pfmdr7* mRNA level in the MEF1 compared with the levels in 3D7 (Fig. 4b, orange), suggesting that MEF1 possibly increases the ability of mefloquine resistance via *pfmdr7* upregulation. In the *pfmdr7* promoter region of MEF1, there were nine mutations could be identified, including deletions and point-mutations, compared with that of 3D7 (Supplementary Fig. 11). We subsequently examined whether those mutations were responsible for the observed *pfmdr7* upregulation using dual luciferases assay on two biologically independent samples (Supplementary Fig. 12). This assay demonstrated that the *pfmdr7* promoter activity in the MEF1 was 2.8–3.2-folds higher than that in the 3D7 (Fig. 4c). In addition, the RT-qPCR analysis of *pfmd7* in transgenic parasites, in which *pfmdr7* was introduced along with the authentic promoters, supported that the *pfmdr7* promoter activity of MEF1 was 3.5-fold higher than that of 3D7

(Supplementary Fig. 13). Therefore, based on these results, we considered that this mutated promoter upregulated *pfmdr7*.

To examine whether basal *pfmdr7* transcription resulted in a similar increase in mefloquine-resistant parasites other than MEF1, we compared the *pfmdr7* mRNA expression levels of between mefloquine-resistant (MEF18 IC$_{50}$: 77.9 ± 8.2 nM, MEF24 IC$_{50}$: 76.6 ± 1.5 nM, Supplementary Data 3 and Supplementary Fig. 14) and sensitive (MEF19 IC$_{50}$: 18.3 ± 1.1 nM, MEF25 IC$_{50}$: 17.6 ± 1.4 nM, and MEF26 IC$_{50}$: 18.7 ± 1.8 nM, Supplementary Data 3 and Supplementary Fig. 14) parasites, which were collected from patients living in the Thai−Myanmar border region (as described for collection of the MEF1). The RT-qPCR analyses of these parasites showed that *pfmdr7* mRNA expression in the mefloquine-resistant parasites, namely MEF18 and 24, was commonly 1.6- to 1.8-fold greater than that in the 3D7 (Fig. 4b). In contrast, the mefloquine-sensitive parasites, namely MEF19, 25, and 26, had *pfmdr7* expression levels that were comparable to or less than those of the 3D7 (Fig. 4b). These results suggested that transcriptional upregulation of *pfmdr7* was a possible factor in the emergence of mefloquine resistance in the field-isolated parasites.

## Discussion

Identification of drug-resistant genes is an essential step for not only understanding resistance mechanisms but also conducting surveillance of the spread of resistant parasites. As examples, the identification of *pfcrt* and *kelch13* have greatly contributed to our understanding of how parasites acquire chloroquine- and artemisinin-resistances, respectively, and such knowledge has substantially improved global surveillance of these resistant parasites[30–35]. In the present study, we demonstrated that a drug-resistance gene could be identified from high-coverage genomic libraries using functional screening. Moreover, we showed that functional screening is a reliable approach for exploring drug-resistant genes; the genomic regions that contained the drug-resistant gene or candidate could be reproducibly identified using the functional screening approach. Such reproducible identification will also be useful for judging whether identified genomic loci will encode drug-resistance genes. Another important feature of our approach is the ability to explore drug-resistant genes in one drug-resistant strain obtained from a single patient. This feature makes our approach additionally useful and more time-efficient than the currently used approach because it often takes years to produce drug-resistant parasites in the laboratory[14–16]. Because of this feature, our approach could be used to identify drug-resistant genes and candidates within a short period after the detection of drug resistance in an endemic area. We anticipate that our functional screening approach will be used to identify the resistance genes related to drugs used in ACT, which will assist in the containment of such genes and limit their spread.

In this study, we transfected the schizonts including 2–4% of full mature forms with the centromere plasmid, pFCENv1, harboring insert DNA fragments of >10-kb in size, resulting in high-coverage genomic libraries. The estimated transfection efficiency using full mature schizont was ~120-fold higher than that of previous methods using DNA-preloads RBCs[36], which was essential for generating high-coverage genomic libraries. Furthermore, this improved transfection efficiency enabled the direct introduction of the insert DNA fragments and pFCENv1 ligation reaction mixture into the parasites. So far, due to the low transfection efficiency of *P. falciparum*, foreign DNA fragments had to be incorporated in the plasmid using *E. coli* prior to the transfection of the parasites. However, several of them would be lost from *E. coli* cells due to their instability caused by the high AT richness[19]. This technical limitation would significantly reduce the complexity and genome coverage of the genomic library. The direct introduction of the ligation reaction mixture allowed for bypassing the *E. coli*-mediated cloning step; it thus solves the technical limitation. However, even though ~500 of the independent parasites were transfected directly, as

shown in this study, this transfection efficiency was considered to be insufficient for generating the genomic library covering the entire genome. Since pFCENv1 can introduce large insert DNA fragments by the function of the centromere, this technical limitation could be solved by its use: the genomic coverage of the library could be increased by introducing larger insert DNA fragments. Taken together, we consider that the modified transfection technique and the centromere plasmid are essential for generating a high-coverage genomic library in *P. falciparum*, and should be used in combination.

The centromere plasmid is maintained as single-copy DNA in the parasite. Therefore, the genes on the delivered genomic DNA fragments are maintained as single-copy genes in the transgenic parasites (Supplementary Figs. 4 and 7). In addition, the promoter activities of most of these delivered genes are likely comparable to those of corresponding genes in the genome of recipient parasites, e.g., 3D7 in this study (Supplementary Fig. 5). Taking these aspects into consideration, the expression levels of multiple genes delivered with pFCENv1 are probably almost identical to their corresponding genes in the recipient genome. As *P. falciparum* increases the CN of drug resistance genes from one to two by gene duplication and could then acquire drug resistance by doubling the gene expression levels, we considered that the expression levels of genes delivered using pFCENv1 are probably sufficient for functional screening.

Loss-of-function (LOF)-type genes confer the drug resistance to cells, such as cancer cells and bacteria, by eliminating or reducing their function[37–39]. In *P. falciparum*, *kelch 13* is considered to be classified into a LOF-type drug resistance gene[40]. Even if LOF-type drug-resistance genes are introduced into the drug-sensitive parasites, such as 3D7, the resulting transgenic parasites would not acquire resistance. Therefore, it is impossible to explore this drug-resistance type via the functional screening proposed in the present study. Another type of drug resistance genes, which called "gain-of-function fashion (GOF)", confer resistance by altering or enhancing gene function[39]. These gene function alternations and enhancements are generally caused by mutation and CN increase. Various drug-resistance genes, such as *pfcrt, pfmdr1, dhfr-ts, and dhps*, identified in *P. falciparum* so far belong to the GOF type of genes[41,42]. As shown by this study, functional screening was used for exploring the GOF type of drug-resistance genes. Taken together, functional screening cannot explore all types of drug-resistance genes, but would be useful for identifying at least the GOF types.

Our findings suggest that *pfmdr7* is a potential mefloquine-resistance gene in *P. falciparum*. ABC transporters are well-characterized membrane proteins with an ATP-binding cassette and are widely conserved in both prokaryotes and eukaryotes. In eukaryotes, ABC transporters export various substrates, such as metabolites, lipids, and sterols, against their concentration gradients using energy from the ATP hydrolysis. In addition, they act as major drug efflux pumps due to their broad substrate specificity. The eukaryotic ABC transporters are classified into seven subfamilies, ABCA−ABCG[43], based on the arrangement of the transmembrane domain and ATP-binding cassette; *pfmdr7* belongs to the ABCB transporter subfamily. As the substrate binding sites of the ABCB transporters are open to the inner leaflets of the lipid bilayers, they are responsible for the export of lipophilic substances, e.g., phosphatidylcholine and bile acid, and can excrete lipophilic drugs, e.g., colchicine[44]. Mefloquine is a lipophilic antimalarial drug[45], it can be thus retained within the parasite cell membranes. Therefore, PfMDR7 might bind to mefloquine within the lipid bilayer and subsequently extrude the drug prior to gaining access to the parasite cytoplasm, potentially resulting in mefloquine resistance.

Increased *pfmdr1* CN, of the ABCB transporter family similar to *pfmdr7*, confers mefloquine resistance in *P. falciparum*. This CN increase results in increased transcripts, conferring mefloquine resistance by enhancing drug efflux from the parasites and/or

sequestration into digestive vacuoles[46]. However, in endemic areas, mefloquine-resistant parasites exist that do not increase the *pfmdr1* CN, such as MEF1, which suggests the involvement of other mefloquine-resistance genes[29]. Our results suggest that *pfmdr7* upregulation could be a factor in mefloquine resistance in such resistant parasites. Furthermore, this upregulation was caused by mutations in the promoter region. Similarly, a recent forward genetic study using progenies between chloroquine-resistant and sensitive *P. vivax* described that *crt* (PVP01_0109300; *pvcrt*) transcriptional upregulation confers drug resistance to the parasite[47]. Since chloroquine resistance in *P. vivax* was not attributed to codon mutations in *pvcrt* in either a study of chloroquine treatment failure in humans and monkeys or in a genome-wide association survey, this *pvcrt* gene upregulation is a possible determinant of chloroquine resistance in this parasite species[48–50]. Therefore, considering the results of this study and the aforementioned previous study on *P. vivax*, transcriptional upregulation of the gene, the products of which can extrude or sequestrate the drugs, might be a way by which *Plasmodium* parasites acquire drug resistance, as well as drug resistance gene CN increase.

## Methods

### Ethical clearance
Ethical clearance for the collection of parasite-infected blood was obtained from the Research Ethics Committees at the Faculty of Medicine, Chiang Mai University (permission number: 187/2554) and from the Department of Medicine, Mie University, Japan (permission number: 1312). This work was conducted in compliance with all relevant ethical standards and regulations governing research involving human samples. Written informed consent was obtained from all patients or the parents or guardians of children. In this study, we did not use any information, such as sex and age, of patients. The information about the parasite strain and patients do not link. Red blood cells (RBCs) were obtained from the Japanese Red Cross (research ID: 25J0143).

### Parasite strains and culture
*Plasmodium falciparum* strains 3D7 and Dd2 were obtained from the Malaria Research and Reference Reagent Resource Centre (http://www.mr4.org). All clinical *P. falciparum* isolates, including the MEF1, were collected from patients living in the Mae Sariang district, Mae Hong Son Province in Thailand according to the abovementioned ethical guidelines. Clonal parasite lines were established from field-isolated parasites and strains 3D7 and Dd2 using limiting dilution, and these lines were used for all studies. All parasite strains were cultivated with human RBCs [type O blood; hematocrit (Ht): 2%] in complete medium, which consisted of RPMI-1640 medium containing 20% AlbuMAX I (Life Technologies), 25-mM HEPES, 0.225% sodium bicarbonate, and 0.38-mM hypoxanthine supplemented with 10-μg/ml gentamicin, and they were incubated under low-oxygen conditions (90% $N_2$, 5% $CO_2$, and 5% $O_2$).

### Transfection of fully mature schizonts
*Plasmodium falciparum* strain 3D7 was used for all transfections. The parasites were roughly synchronized using a 5% sorbitol treatment prior to tight synchronization. When most of the parasites had developed into schizonts, they were purified using a 40%–70% discontinuous Percoll gradient solution (GE Healthcare Life Sciences) with 6% sorbitol. Purified schizonts were cultured with fresh RBCs for 4 h and then treated with 5% sorbitol. The resulting parasites were synchronized within a window of ~4 h. These Percoll and sorbitol synchronizations were repeated three times, resulting in tightly synchronized parasites. The emergence of fully mature schizonts was microscopically monitored from 88 h after the final synchronization, and their ratios were compared with the total schizont numbers every 2 h. When the ratio of fully mature schizonts reached the maximum

level, the parasites were purified using a discontinuous Percoll gradient. Purified schizonts consisted of both immature and fully mature forms, and the ratio of fully mature schizonts usually reached ~1%–2%. The DNA samples were dissolved in 100 μl of Parasite Nucleofector II solution (LONZA), after which they were mixed with the purified schizonts ($1.0 \times 10^8$). The parasites were transfected using the U-033 program on a Nucleofector II device (LONZA). Immediately after electroporation, the transfected parasites were mixed with 0.1 ml of complete medium and then cultured in 5 ml of complete medium with fresh RBCs. Using pyrimethamine, drug selection of transgenic parasites was initiated 60 h after transfection. This transfection procedure using fully mature schizonts has been already published[21].

### Determination of transfection efficiency
To evaluate transfection efficiency, the number of independently transfected parasites (the iTP number) was estimated based on the percentage of parasitemia and the multiplication rate of the transgenic parasite. The multiplication rate (3.7 per cell cycle) of the transgenic parasite with the introduced pFCENv1 was determined in the presence of pyrimethamine[20]. The iTP number was calculated using the following equation[22,23]:

$$T \times P/100 = [I \times (3.7)^{D/2}],$$

where $T$ is the total number of RBCs in culture, which comprised 5 ml of medium with 2% Ht; $D$ is the days after transfection; $P$ is the percentage of parasitemia on day $D$; and $I$ is the iTP number.

In addition, the iTP number was estimated directly. Briefly, 1/10 and $1/10^2$ of the transfected parasites were resuspended to 50 ml of complete medium with fresh human RBCs immediately after transfection; 48 ml of each resuspended culture was then dispensed into 48 wells (1 ml per well). The emergence of transgenic parasites in each well was detected in the presence of pyrimethamine during long-term culture. This assay was repeated 10 times. The iTP number was estimated by calculating using the counted number of parasite-positive wells in 10 assays.

### Genomic library construction
The genomic DNA of drug-resistant parasites, i.e., strains Dd2 and MEF1, was partially digested using *Sau*3AI and separated using electrophoresis on a 0.75% low-melting-point agarose gel (LONZA). The gel sections containing >10 kb of DNA fragments were excised, melted by heating, and then digested with thermostable β-agarase (NIPPON GENE). After digestion, the DNA fragments were extracted using phenol/chloroform/isoamyl alcohol (PCI; 25:24:1, v/v) and then precipitated with 100% ethanol. The pFCENv1 was completely digested using *Bam*HI and dephosphorylated using alkaline phosphatase (Roche). Subsequently, 5 μg of *Bam*HI-digested pFCENv1 was ligated using the same amount of partially digested genomic DNA, after which it was purified using PCI and precipitated using ethanol. To generate the genomic library of the Dd2, we prepared five sets of ligated samples and separately introduced them into the 3D7 by repeating the electroporation five times using the Nucleofector II device, as described above. All transfected parasites were assembled to yield the library. This library construction experiment was repeated three times, thereby yielding three genomic libraries of the Dd2. To generate the library of the MEF1, three sets of ligated samples were separately introduced into parasites by repeating the electroporation three times. All transfected parasites were assembled. This library construction experiment was repeated eight times in total, thereby yielding eight genomic libraries of the MEF1 were used for the first experiment to identify a mefloquine-resistance gene. An additional eight genomic libraries were generated in the manner described above, and these were used for the second experiment. Pyrimethamine treatment of transgenic parasites in the library was

initiated 60 h after electroporation, and the resulting parasites were stored at −80 °C. The genomic coverage of the constructed genomic library was estimated using the following equation:

$$C = (I \times L)/G,$$

where $C$ is the coverage of the genomic library, I is the iTP number generated in one experiment, $L$ is the average length of the inserted DNA fragments, and $G$ is the total length (25 Mb) of the *P. falciparum* genome.

### CHEF electrophoresis and Southern blot hybridization

The infected RBCs were collected and then lysed with RBC-L buffer, which consisted of 0.15 M $NH_4Cl$, 10 mM $KHCO_3$, and 1 mM EDTA (pH 8.0). After RBC lysis, the parasites were collected using centrifugation at 590×$g$ for 6 min. The obtained parasites were washed with phosphate-buffered saline and mixed with 2% low-melting-point agarose at a ratio of 1:1 (v/v). The agarose blocks containing parasites were soaked in sarcosyl-EDTA buffer (comprising 0.5 M EDTA at pH 8.0 and 1% sarcosyl) and then treated with proteinase K (100 μg/ml) at 37 °C overnight. The agarose blocks were placed in a 1.5% agarose gel, and the pFCENv1 in which the genomic DNA fragments (>10 kb) were inserted was separated from the parasite chromosome using CHEF electrophoresis under the following conditions: initial switching time: 1.0 s; final switching time: 6.0 s; angle: 120°; voltage gradient: 6 V/cm; run time: 20 h; and temperature: 14 °C. After electrophoresis, the DNA was blotted onto a nitrocellulose membrane, and the pFCENv1 containing the insert DNA was detected using Southern blot hybridization with the digoxigenin-labeled human dihydrofolate reductase (*hdhfr*) gene employed as a DNA probe. Hybridized signals were detected using a LAS-3000 Mini Lumino-Image Analyzer (FUJIFILM). The lengths of the inserted DNA fragments were estimated based on the mobility of the detected signals compared with those of the 8–48 kb DNA marker (Bio-Rad).

### Functional screening of a drug-resistant parasite from the library

The transgenic parasites with newly acquired chloroquine resistance were selected from the genomic library using three rounds of drug screening. Briefly, the parasites in the genomic library of the Dd2 were first treated with 20 nM chloroquine for 4 days and subsequently cultured until surviving parasites were detected. These were treated again with the same concentration of chloroquine for 4 days and cultured until parasite survival was again detected. The parasites that survived these two rounds of chloroquine screening were further treated with 20 nM chloroquine for 6 days. The chloroquine-resistance levels of the surviving parasites after three rounds of screening were confirmed by treatment with 20 nM chloroquine for 6 days.

The parasites that acquired mefloquine resistance were also screened from the genomic library of the MEF1 using three rounds of mefloquine screening. The parasites were treated with 15 nM mefloquine for 4 days in the first and second rounds and were further treated for 6 days in the third round. The surviving parasites obtained after three rounds of screening were further treated with 15 nM mefloquine for 6 days. Finally, the mefloquine resistance of the surviving parasites was confirmed.

### Sequence analysis of the ends of the DNA fragments inserted into pFCENv1

Sequence analysis of the ends of the insert DNA fragments was performed using the GenomeWalker Universal Kit according to the manufacturer's protocol but with some modifications (TAKARA BIO). The pFCENv1 containing the insert DNA fragments was recovered from the clonal parasites, which were established from the surviving parasites after three rounds of chloroquine or mefloquine screening. The

purified DNA samples were digested using *Ssp*I and then ligated to the DNA adaptors, which were supplied in the kit, as described above. PCR was performed using the adaptor-ligated DNA as a template with the following primers specific to the adaptor and pFCENv1: primers specific to pFCENv1, 5′-CTTCTGCCAGAATACCCAGGTGTTCTCTC-3′, and 5′-GTATTGGGAATTCCAGCACACTGGCGGC-3′; a primer for the adaptor sequence, 5′-GTAATACGACTCACTATAGGGCACG-3′. Subsequently, nested PCRs were performed using the product of the first PCR as a template and another set of primers specific to the adaptor and pFCENv1 as follows: primers specific to pFCENv1, 5′-CCAGCAC ACTGGCGGCCGTTACTAG-3′ and 5′-GAATGATTAGTCGAGGGATA TGGCAGC-3′; a primer for the adaptor sequence, 5′-ACTATAGG GCACGCGTGGTC-3′. Primers specific to pFCENv1 were designed at both sides of the cloning site, i.e., the *Bam*HI restriction site on this plasmid. The sequence similarities of amplified products were searched using BLAST and the genomic sequence of the 3D7 as a reference sequence. The genomic regions encoded by the inserted DNA fragments in pFCENv1 were identified based on the BLAST results.

### Measurement of the IC$_{50}$ values of chloroquine and mefloquine

Prior to measuring the IC$_{50}$ values, the parasites were synchronized twice using 5% sorbitol. When they were mostly at the ring stage, the percentage of parasitemia was adjusted to 0.5%, and 50 μl of each parasite culture was dispensed into a 96-well plate. A stock solution of chloroquine was serially diluted in a complete medium, and 50 μl of each diluted drug medium was dispensed in quadruplicate into test wells containing the parasites. The final concentration of chloroquine was 0–800 nM. After 144 h of incubation, 0.02 μl of SYBR Green I (Invitrogen) was diluted 5000-fold in lysis buffer (comprising 20 mM Tris-HCl at pH 7.5, 5 mM EDTA, 0.008% of saponin, and 0.08% Triton X-100) and then added to each test well. The plates were shaken until all RBCs were completely lysed, after which they were placed at room temperature in the dark for 1 h. SYBR Green I fluorescence was measured using a Filter Max F5 microplate reader (Molecular Devices) with excitation and emission wavelengths at 485 and 530 nm, respectively. In the assay used to measure the IC$_{50}$ value of mefloquine, the final drug concentration was 0–100 nM, and the parasites were cultured for 72 h before the addition of SYBR Green I solution. The IC$_{50}$ values were calculated based on the detected fluorescence intensities using GraphPad Prism 6 software. Briefly, the survival rates were plotted against the logarithm of the drug concentration, and the curve fittings were performed using nonlinear regression to yield the IC$_{50}$, i.e., the drug concentration that produced a 50% survival rate. In this curve-fitting procedure, we selected the "log (inhibitor) vs. response (variable slope)" option built into the software according to the manufacturer's instructions.

### CN analysis of the genes on pFCENv1, which were recovered from parasite clones selected by functional screening

The CN of the genes on pFCENv1 in parasite clones, selected from libraries via chloroquine- and mefloquine- screening, were examined using qPCR. The genomic DNA (0.5 ng) purified from those selected parasite clones was used as template DNA and qPCR analyses were performed in quadruplicates. The β-tubulin (PF3D7_1008700) and fructose-bisphosphate aldolase (PF3D7_1444800) were used as single-copy reference genes. To examine the *pfcrt* CN on the pFCENv1 in parasite clones, obtained from dd2-lib1 and -lib2, we used primers specific to *pfcrt* gene. In addition, the *pfmdr7* CN on pFCENv1 in parasite clones from mef-lib3 and -lib6 were analyzed by using the primer specific to *pfmdr7*. The primers used in this assay were as follows: *pfcrt*, 5′- GTTCTTGTAAGACCTATGAAGGC-3′ and 5′-TTTAG-GAACGACACCGAAGC-3′; *pfmdr7*, 5′-TGCGGTGTTACGATCTATTGT GG-3′ and 5′-ACATCAAATTCTCACGCACAGTC-3′; PF3D7_1008700, 5′-

CTATGGATAGTGTTCGTGCTGG-3′ and 5′-CTGCATCTATCAATTCAG-CACC-3′; and PF3D7_1444800, 5′-TGTACCACCAGCCTTACCAG-3′ and 5′-TTCCTTGCCATGTGTTCAAT-3′.

## Quantitative reverse transcription PCR for mRNA expression of *pfmdr7* and *pfcrt*

To quantify *pfmdr7* transcripts, total RNA was purified from parasites 30 h after infection when the expression of *pfmdr7* had reached the maximum level (http://plasmodb.org/plasmo/). Total RNA was purified from four biologically independent samples for each parasite strain, i.e., the strains 3D7 and MEF1 and both transgenic parasites into which the *pfmdr7* genes of strains 3D7 and MEF1 were introduced. cDNAs were synthesized from 0.5 μg of each purified total RNA sample using a PrimeScript RT Reagent Kit with a gDNA eraser according to the manufacturer's instructions (TAKARA). Subsequently, RT-qPCR of *pfmdr7* was performed using SYBR Premix DimerEraser (TAKARA) and a Thermal Cycler Dice Real-time System (TAKARA). The fructose-bisphosphate aldolase gene (PF3D7_1444800) was used as an internal control[51]. The *pfmdr7* mRNA relative expression levels were calculated based on the CT values using the $2^{-\Delta\Delta CT}$ method. In addition, the mRNA expression of *pfmdr7* in field-isolated mefloquine-resistant and -sensitive parasites was quantified in a similar manner to that described above. All assays were performed in more than triplicates.

To compare the promoter activities of *pfcrt* between strains 3D7 and Dd2, we performed qPCR analysis using the RNA isolated from those strains. Total RNA were isolated from parasites 6 h after infection as the *pcrt* expressed maximally ~10 h after infection (http://plasmodb.org/plasmo/). The RT-qPCR experiments of *pfcrt* were performed in a similar manner as those of *pfmdr7*. The primers used in all these analyses were same as those used in the above-described CN analysis. All assays were performed in quadruplicates.

## CN variation analysis of the *pfmdr7* gene in strain MEF1

The CN of *pfmdr7* was determined based on the whole-genome sequencing data obtained for the MEF1 using an Ion-proton System (Applied Biosystems) and the read-depth approach[52]. Briefly, the obtained reads were mapped to the genomic sequence as well as the gene-coding sequence of the 3D7, both of which are available in PlasmoDB (http://plasmodb.org/plasmo/). After duplicate reads were removed using the "rmdup" option in SAMtools[53] (version 1.10), the average coverage within nonoverlapping 50 bp windows was calculated throughout the genomic and coding regions of *pfmdr7*; subsequently, the maximum coverage (Max_target) was determined. Similarly, the maximum coverage of eight known single-copy internal-control genes (Supplementary Data 1) was determined, and the averages (AveMax_control) were calculated. The *pfmdr7* CN was determined based on the ratio of Max_target to AveMax_control. We also determined the CNs of all the ABC transporters of *P. falciparum* (Supplementary Data 1). The genomic sequence data for the MEF1 is deposited at DDBJ (http://www.ddbj.nig.ac.jp/index-j.html; accession number: SAMD00047017).

## Assay for the promoter activity using two luciferases

We compared the MEF1 and 3D7 *pfmdr7* promoter activities using firefly (*Photinus pyralis*) and sea pansy (*Renilla reniformis*) luciferases according to the manufacturer's protocol (Promega). The promoters of *pfmdr7* were amplified from MEF1 and 3D7 using the primer set; 5′-CTCACTATAGGGCGATTGGGTACCGTGCTCACCCAAGTTTATCATGA-CAAAATTAGG-3′, and 5′-CATCTTCCATCTCGACGCTAGCTACGTATG-TATTATGTTG-3′. The amplified PCR products were cloned 5′-upstream of the firefly luciferase gene on the pfLuc2 plasmid (Supplementary Fig. 12). The sea pansy luciferase gene was transcribed using the promoter of *P. berghei* elongation factor 1α and used as an internal control. The resulting plasmids were introduced into the 3D7

via electroporation and the transgenic parasites with the plasmids were selected using pyrimethamine. The transfection was performed in duplicates for each plasmid with the *pfmdr7* promoters of MEF1 and 3D7. The luciferase activities of firefly and sea pansy luciferases in transgenic parasites were measured in triplicates using beetle luciferin and coelenteramide, respectively, as substrates, according to the manufacturer's protocol. The statistical differences between the obtained results were evaluated using the Student's *t* test. All assays were performed in triplicates.

## Functional analyses of three genes encoded by the genomic DNA fragments identified using functional screening

The genomic DNA fragments encoding *pfmdr7* and its 5′-upstream sequence from strains 3D7 and MEF1 were amplified using PCR and the following primer set: 5′-AAAGGTACCGTGATTCATAAGAGTGGCTAT TCC-3′ and 5′-AAACTCGAGTTATTCACTCACACCTTGTGTAAG-3′. The amplified products were cloned using *E. coli* into *Kpn*I-*Xho*I sites in the modified pFCENv1, in which multicloning sites and the 3′-UTR of heat-shock protein 70 (PbHSP70, PBANKA_0711900) of *P. berghei* were placed upstream of the centromeric sequence of *P. falciparum*. The resulting plasmids were introduced into schizonts of the 3D7 in triplicate. Using pyrimethamine, drug selection of transgenic parasites was initiated 60 h after transfection, and clonal transgenic parasites were obtained using limiting dilution. In addition, genomic DNA fragments encoding *atp11* and *rpl1* were amplified using the following primer set: *atp11*, 5′-AAAGGTACCAAACATTTTTAACCCTTTCAAA AATAAACC-3′ and 5′-AAACTCGAGTTTATCATATAATATCATTTAAAA-TACACAG-3′; *rpl1*, 5′-AAAGGTACCCACAATTTAGAAGACAATCCA-TAAG-3′ and 5′-AAACTCGAGTTGCACATAAAAATATGAGAGCTGCTC-3′. Because these fragments could not be cloned into pFCENv1 using *E. coli*, they were directly introduced into the parasites via electroporation. Briefly, the amplified DNA fragments (2.5 μg) encoding the aforementioned genes were ligated at *Kpn*I-*Xho*I sites in the modified pFCENv1 and then directly introduced into parasites in duplicates. The clonal parasites into which pFCENv1 with each DNA fragment was introduced, were established using limiting dilution. Subsequently, the DNA fragments inserted into pFCENv1 were amplified using genomic DNA recovered from the clonal parasites as templates, the specific primer for pFCENv1, and the *Xho*I primers used for amplification of each gene. Subsequently, the sequences of the *atp11* and *rpl1* genes inserted in the plasmid were confirmed. All of the resultant transgenic parasites in which the *pfmdr7*, *atp11*, and *rpl1* genes were introduced, were treated with 15 nM mefloquine for 6 days, and the mefloquine resistance of these parasites was evaluated based on their survival.

### Reporting summary

Further information on research design is available in the Nature Research Reporting Summary linked to this article.

## Data availability

The genomic sequencing data generated in this study have been deposited in the NCBI database under accession code DRX049711 [https://www.ncbi.nlm.nih.gov/sra/DRX049711%5baccn%5d]. In addition, the genomic sequence of *P. falciparum* 3D7 version 35, which is deposited in PlasmoDB (https://plasmodb.org/plasmo/app/), was used as the reference sequence. The remaining data are available within the article, supplementary information, or source data file. Source data file is provided with this paper. Source data are provided with this paper.

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

## Acknowledgements

All studies were supported by Grants-in-Aid for Scientific Research (20H03477 and 19K22527 to S.I., 18K07084, and 21K06985 to N.S., and 17H01542 to M.Y.), which were funded by Japan Society for the Promotion of Science, and by the Japan Agency for Medical Research and Development (AMED) under grant numbers 21jm0210061h0004 (to S.I.), 21wm0325018 (to N.S.), and 21wm0225014 (to N.S.).

## Author contributions

S.I., C.U., and D.S. conceived the study, performed all experiments, and wrote the manuscript. M.Y. wrote the manuscript together with S.I. T.N. and performed the whole-genome sequencing of the MEF1. N.S. and R.K. calculated the CN of *PfMDR7*. C.U. conducted sample collection from patients. S.K. and S.S. established the field-isolated parasites together with C.U.

## Competing interests

The authors declare no competing interests.
