## [Peer review file · Nature Communications]

REVIEWER COMMENTS

Reviewer #1 (Remarks to the Author):

In this manuscript the authors carried out transgenesis-based screenings for putative new drug resistance genes in human malaria parasites *Plasmodium falciparum*. The main purpose of this manuscript was to derive a technique by which new drug resistance marker can be identified by large scale transgenic screens with genomic libraries of resistance strains. Indeed, drug resistance in malaria is a highly important issue and identifying new effector genes and as such genetic markers is of a high interest. As such this manuscript addresses an important issue and is timely.

Unfortunately, in spite of this, in my view, the results in this manuscript are not convincing about neither the derived transgenic technique nor the identified genes to warrant a high impact publication in Nature Communications. There might be a possibility that with a considerable rearrangements this manuscript could be still published in NC, if certain concerns are cleared (below). In my view that would require a complete redo and thus new submission.

Here are some specific concerns:

1. In the first part of the manuscript the authors claim to achieve a high throughput transgenesis with genetic library of *P. falciparum* with >10kb fragments. The authors estimated that ~700-500 parasites have been transfected each of the three times this experiment was conducted. It is highly unclear how was this estimation done. All that the authors show is 10 Southern blots with positive signals indicating the presence of the plasmid. In my view much stronger evidence should be provided for this rather a strong statement.

2. If the estimation holds true, it is unclear to me what technological advances were implemented in this study compared to all other transfection approaches in *P. falciparum* which are typically tedious and low efficiency. From the current version of the manuscript, I understand that the authors used just the standard approach. On one side this shows a huge labor intensive effort for which the authors deserve a credit, however, as such this study brings no new technological advancements. As such this technique of genomic library transgenesis does not seem to be very operational for future studies. I might have missed it; but are there any technological improvements presented in this work, outside of doing more of the same?

3. Identifying *pfcr*t in the chloroquine screen also in my view does not validate this approach entirely. Indeed, *pfcr*t gene is well known to drive chloroquine resistance, hence, identifying it in this library screen is a good sign. However, as I understand the approach, the selected chloroquine resistant clone

was expressing the DD2 resistant (mutated) allele of *pfCRT* in a dominant negative fashion (not discussed by author). Essentially, the mutated allele was expressed along the indigenous allele of the *P. falciparum* strain used for the transfection which was sufficient to drive resistance. This by itself is interesting but this indicates that in order for to this assay to identify other markers of resistance, these must also work in as dominant negative. Hence, many other genes possibly involved in resistance will be missed by this approach so could not be excluded. This is a huge limitation.

4. Related to the point 3. As I understand it, the genetic library is constructed such that each transgenic gene is expressed from its original promoter. Essentially, this approach is designed to create an artificial CNVs by introducing an additional copy of a gene expressed episomally. Unfortunately such approach does not automatically guarantee expression of overexpression. Once again, this could lead to huge amounts of false negative results which render this approach not very impactful. It would be helpful to show how many genes/proteins were indeed overexpressed in the constructed libraries.

5. This issue is relevant for the subsequent part of the study in which *pfmdr7* gene was identified as a mefloquine resistance marker. In my view this result is somewhat misleading. Although the used the mefloquine resistant strain for the genetic library contraction, the obtained result does not explain its resistance mechanism. Instead, it shows that an artificially increased copy number of *pfmdr7* could lead to mefloquine resistance in any strain. Although that is interesting, the relevance of this observation for clinical setting is unclear at best. So far no clinical or epidemiological evidence was observed form *pfmdr7* involvement in drug resistance of *P. falciparum* and the authors di not try to find it. As such this could be a highly artificial observation of an *in vitro* phenotype. Also, this could be a lucky situation in which an increased copy number of *pfmdr7* gene drives resistance. This might not be the case for other possibly relevant markers that will be missed by this technique. Given how hard and labor intensive these experimental approach seems to be, this would not be my method of choice to search for genetic markers.

Reviewer #2 (Remarks to the Author):

The approach described and validated in this manuscript constitutes a significant advance for the malaria field, providing a new means for the identification of drug resistance mechanisms in *Plasmodium falciparum*. While perhaps not novel in concept, given that the authors have validated the use of a similar approach in the genetically tractable murine malaria parasite *Plasmodium berghei* (Iwanaga et al. 2012 Genome Res), its adaptation for use with the most deadly human malaria parasite, *P. falciparum*, is important. For some antimalarial lead compounds, it is possible to generate resistant parasites readily *in vitro* in a matter of weeks and then sequence their genomes to determine the genetic basis for resistance. However, for other antimalarials and lead compounds, this is not possible. A

low propensity for resistance development, and an inability to generate resistance readily in vitro, bodes well for the longevity of an antimalarial, and it is arguably such compounds that should be prioritised for clinical development. The development in this study of a time-efficient approach to study resistance mechanisms for such compounds, that arise in the field or during the course of clinical trials, is extremely valuable.

The authors used their new approach to obtain interesting, novel data on mechanisms of mefloquine resistance. They identified *pfmdr7* as a resistance determinant for this clinically used drug. Interestingly, through well designed experiments, the authors provided evidence that transcriptional upregulation of *pfmdr7* (rather than the more common finding of copy number variation or SNPs) was the likely mechanism of mefloquine resistance in a clinical isolate. Such a resistance mechanism would not have been discovered with a whole genome sequencing approach, highlighting an additional major advantage of the authors' new approach to the identification of drug resistance genes.

The methodology is sound and well described. This reviewer only noted one potential weakness – that statistical analyses were not performed to compare the IC₅₀ values obtained in parasite proliferation assays for chloroquine or mefloquine between different parasite lines, possibly because independent biological replicates of these experiments were not performed. It is not clear whether the data from parasite proliferation assays shown in the Figures were derived from a single experiment or more. Ideally, multiple independent experiments should be performed for all parasite proliferation assays, such that IC₅₀ values can be provided as means with error values, and differences between parasite lines can be tested for statistical significance. It is noted however that the ability of parasites to survive in the presence of the drug provided complementary data enabling sensitive and resistant parasites to be distinguished from one another. At the minimum, the authors should state the number of biological and/or technical replicates from which the parasite proliferation data were obtained, in the relevant Figure legends.

On the whole, the manuscript is well written and the Figures are well presented. The methods are described in detail, which will allow similar experiments to be undertaken by other groups. Some minor suggestions for further improving clarity are outlined below:

1. the concentration of pyrimethamine used to select for transfected parasites should be stated
2. '> 10 kb of DNA fragments' or similar (used on two occasions) is not clear – consider replacing with 'DNA fragments > 10 kb in size'
3. line 117: it is not clear whether the sizes referred to on this line include the vector size or not
4. the phrase 'A clonal parasite was established from each surviving parasite' (used on two occasions) is not clear – consider replacing with 'A clonal parasite line was established from each culture containing surviving parasites'
5. lines 174-175: the data provided for the IC₅₀s and fold difference appear to contradict one another

6. line 293: should this be 'efflux from parasites and/or sequestration into the digestive vacuole', given what is known about the localisation of PfMDR1?
7. line 522: there is a word missing - L is the average [length?]
8. Supp. Fig. 7: is Day 0 when the drug was first added? It would be good to clarify.

Reviewer #3 (Remarks to the Author):

This manuscript describes a potentially powerful approach to identifying the gene(s) underlying drug resistance in the human malaria parasite. The system includes generating a resistant parasite genomic library with large (>10 kb) inserts in an artificial chromosome vector. The library will be directly transfected to a sensitive lab parasite strain (in their case, 3D7). The authors have shown that they could obtain an average of ~500 transfectants with an average insert size of 25 kb, roughly covering half of the genome. Two repeated transfections would cover one genome equivalence. These library of transfectants are subjected to selection with the drug of interest, and emerging clones can be genotyped to determine the insert genes that might be responsible for the resistance phenotype.

As a proof of principle, they have demonstrated that this approach works for determining pfprt as the determinant of chloroquine resistance. Then they used a field parasite isolate from western Thailand showing low-level resistance to mefloquine (MQ) to identify that *mdr7* was responsible for the resistance to MQ.

Major comment:

I agree with the authors that this will be a very useful functional method to identify drug resistance genes. However, I also feel that the MQ resistance part is a bit weak, and deserves improvement. The identification of 4 clones with overlapping chromosome 12 fragments is quite convincing, but the subsequent experiments did not substantially corroborate the data that *mdr7* is the responsible gene. 1) It did not show the reason why *mdr7* had higher expression in the resistant strains? They did not identify mutations in the ORF, but what about the promoter region? For the sequence introduced into 3D7, does it contain the promoter sequence? If yes, is it different from 3D7? 2) For the *mdr7* gene introduced into 3D7, is the extra copy from the artificial chromosome responsible for the overexpression? If so, I would assume that inserting an extra copy of the 3D7 *mdr7* gene might have the same effect. 3) The 1.6x increase in *mdr7* gene expression is weak evidence supporting the conclusion. I wonder whether the authors could express *mdr7* ORF using different promoters (with different strengths) in their artificial chromosome to further demonstrate this?

Minor comments:

1. In the second paragraph, the authors stated the lengthy process of identifying k13 as the gene mediating artemisinin resistance and the desire for an alternative approach to speed up this process. I think this point needs to be discussed, since the functional screening approach described here may not be suitable for identifying artemisinin resistance genes given the constant drug pressure needed to screen the transfectant library.

2. Line 100: The manuscript style presents results first. It would make it easier to understand what was used as the target parasite without going to the method section in the back (in this case, *P. falciparum* 3D7).

3. Line 102: explain why pyrimethamine selection (dhfr cassette in the artificial chromosome vector?)

4. Paragraph starting at line 210: it is difficult to understand how this was done. Which part of each gene was cloned and expressed? Does that include the promoter and 3' region? This information is critical for the readers to understand how this would increase *mdr7* expression?

5. Some comments on validation using this approach would be nice. Since the putative gene might be large (for example, *mdr7* ORF plus promoter/3' region may exceed 5-6 kb), amplifying the large piece and inserting it in the vector might not be easy. Thus, this caveat may be mentioned, given that many plasmodium genes are large.

Reviewer #1 (Remarks to the Author):

To reviewer #1,

We greatly appreciate the important comments and meaningful suggestions of the Reviewer. We found them extremely useful for revising and significantly improving our manuscript, compared with the previous version. Please find our point-by-point responses below.

1. In the first part of the manuscript the authors claim to achieve a high throughput transgenesis with genetic library of P. falciparum with >10kb fragments. The authors estimated that ~700-500 parasites have been transfected each of the three times this experiment was conducted. It is highly unclear how was this estimation done. All that the authors show is 10 Southern blots with positive signals indicating the presence of the plasmid. In my view much stronger evidence should be provided for this rather a strong statement.

(Response 1)

The number of independently transfected parasites (iTP) was estimated according to the method described by Andrew P Waters and Janse Chris (Janse, C. J. *et al. Mol. Biochem. Parasitol.* **145**, 60–70 (2006) and Janse, C. J., Ramesar, J. & Waters, A. P. *Nat. Protoc.* **1**, 346–356 (2006)). The reported equation was rearranged and used in this study as follows:

$$T \times P/100 = [I \times (3.7)^{D/2}],$$

where T indicates the total number of RBCs in culture, comprising 5 ml of medium with 2% Ht; D indicates the days after transfection; P indicates the percentage of parasitemia on day D; and I indicates the iTP number. This equation was described in the Materials and Methods section of the previous manuscript and not in the Results section. In addition, we did not cite those two important references correctly. We briefly described the estimation principle in the Results section of the revised manuscript (P.6. I.112-I.114), cited the references appropriately (ref22, 23), and described the details in the Materials and Methods section (P.27. I.615-I.624). The iTP number was estimated as 561.7 using this method. Using a different assay, we further verified whether the estimated iTP number is approximately correct. To make this assay easy to understand, we included a new Supplementary Figure S1b, with a schematic drawing and the obtained results. Briefly, the 1/10 and 1/10² of the culture containing transfected parasites were dispensed into 48 wells immediately after electroporation; the number of parasite-positive wells was counted after long-term cultivation; and the number of iTP was estimated by back calculating the counted number of parasite-positive wells (P.6. I.119-I.125, P.27. I.625- P.28I.631). The average iTP number was estimated to be approximately 400 in ten assays. (Supplementary Fig. 1b). The results obtained using two different methods were consistent. Therefore, taken these results into consideration, we estimated that at least 500 parasite clones were transfected independently in the present experiment.

2. If the estimation holds true, it is unclear to me what technological advances were implemented in this study compared to all other transfection approaches in *P. falciparum* which are typically tedious and low efficiency. From the current version of the manuscript, I understand that the authors used just the standard approach. On one side this shows a huge labor intensive effort for which the authors deserve a credit, however, as such this study brings no new technological advancements. As such this technique of genomic library transgenesis does not seem to be very operational for future studies. I might have missed it; but are there any technological improvements presented in this work, outside of doing more of the same?

(Response 2)

We believe that the technical advancement of this study is that a genomic library that covers the entire *P. falciparum* genome could be generated. To the best of our knowledge, such gene library has never been generated for *P. falciparum*. This success allows the genome-wide functional screening of drug resistance genes. This technical advancement is achieved using the modified transfection protocol and the centromere plasmid, as explained below.

Currently, the transfection approach using schizonts is a standard method and is used in several studies. We recently published the modified and improved protocol in *Scientific Reports* (Nishi, T., Shinzawa, N., Yuda, M. & Iwanaga, S. *Sci. Rep.* **11**, 1–12 (2021)) and used it in this study. Briefly, we enriched the fully mature schizonts, which have lost either RBC or PV membranes, and used them for transfection. Approximately $2.0\text{--}3.0 \times 10^4$ parasites could be transfected independently via this protocol, using 50 μg of the centromere plasmid. We consider that this high efficiency is essential for generating high-coverage genomic library.

To date, a DNA fragment has always been incorporated in a plasmid using *E. coli* before parasite transfection due to the extremely low transfection efficiency of *P. falciparum*. However, it could not be used for the library construction of *P. falciparum* in the present study, as it is impossible to incorporate the genomic DNA fragments in the plasmid using *E. coli* due to their size (>10 kb) and AT richness. Due to this technical limitation, it has been difficult to generate the genomic library covering the entire genome. The improved transfection protocol allows the use of the ligation reaction mixture of pFCENV1 and genomic DNA fragments directly for transfection; thus, we could bypass the cloning step using *E. coli*. Therefore, the technical limitation can be solved by such direct introduction of genomic DNA fragments into the parasite. Owing to these two advantages, the use of this modified transfection protocol is key to successful genomic library generation in this study.

However, even though several hundreds of parasites can be transfected using our protocol, this efficiency is still one or two order lower than that of the protocol used for the rodent malaria parasite *P. berghei* (Janse, C. J., Ramesar, J. & Waters, A. P. *Nat. Protoc.* **1**, 346–356 (2006)). The genomic coverage

of the library can be increased using large DNA fragments according to the following equation (P.29. l.656-l.661):

$$C = (l \times L)/G,$$

where C, l, L, and G indicate the coverage of the genomic library, number of independently transfected parasites, average length of the inserted DNA fragments, and total length (25 Mb) of the *P. falciparum* genome, respectively. To introduce the large DNA fragment into the parasites, the centromere plasmid should be used. As shown in the present results, we indeed introduced the long DNA fragment stably in the parasites using the centromere plasmid and generated the genomic library, thus covering their entire genome. Therefore, we consider that using the centromere plasmid is the second key to success.

This technical advancement will not only be useful for exploring drug resistance genes, as shown in this study, but also screening genes of any interesting function, such as interaction with host molecules. Therefore, we hope that our functional screening approach will be a potential option for identifying genes in various molecular genetic studies.

In the previous version of the manuscript, we did not describe the modified transfection protocol in detail or cited the references. In the revised manuscript, we provided these details in the Materials and Methods (P.26. l. 593-l.612) and Results sections (P.5. l.101-105, ref21), respectively. In addition, we described the advantage of the centromere plasmid for generating high-coverage genome library in the Discussion section (P.16. l.350-358).

3. Identifying pfcr1 in the chloroquine screen also in my view does not validate this approach entirely. Indeed, pfcr1 gene is well known to drive chloroquine resistance, hence, identifying it in this library screen is a good sign. However, as I understand the approach, the selected chloroquine resistant clone was expressing the DD2 resistant (mutated) allele of pfcr1 in a dominant negative fashion (not discussed by author). Essentially, the mutated allele was expressed along the indigenous allele of the P. falciparum strain used for the transfection which was sufficient to drive resistance. This by itself is interesting but this indicates that in order for to this assay to identify other markers of resistance, these must also work in as dominant negative. Hence, many other genes possibly involved in resistance will be missed by this approach so could not be excluded. This is a huge limitation.

(Response 3)

We appreciate the important suggestion of the Reviewer. In general, the drug resistance genes are classified into two categories: gain-of-function and the loss-of-function. Our approach cannot be used for screening the loss-of-function drug resistance genes. Even if such drug resistance genes are introduced into the wild-type parasite, the resulting transgenic parasite would never acquire any resistance. Therefore, it is impossible to screen such drug resistance genes functionally. However, as shown in this study, our approach is useful for screening drug resistance genes, which confer additional or new

functions to the parasites via mutation, copy number increase, and upregulated expression (“gain-of-function”), with the involvement of PfCRT, PfMDR1, DHFR-TS, DHPS, and PfMDR7. So far, in *Plasmodium* spp, most identified drug resistance genes confer resistance through gain-of-function. Therefore, we considered that our functional screening approach cannot explore all drug resistance genes but would be useful for screening at least the gain-of-function type of drug resistance genes. We described this limitation of our approach in the Discussion section of the revised manuscript (P.16. l.370- P.17.l.383).

In our opinion, the expression “dominant negative” is generally used when the functions of the mutant protein are more dominant than those of the wild-type protein (or drug resistant molecule) and when the overexpression of the mutant protein impairs the function of the wild type protein (or drug resistant molecule). To the best of our knowledge, no study has yet reported that the cells including parasites would acquire drug resistance due to the dominant negative drug resistance genes. In contrast, certain studies have described that drug resistance becomes weak upon dominant negative inhibition by mutant proteins (Int J Cancer. 2002 Aug 10;100(5):542-8. doi: 10.1002/ijc.10526). We might have misunderstood the comment of the Reviewer regarding the term “dominant negative fashion.” If further explanation is required concerning the drug resistance gene that were not identified using our method, we request the Reviewer to explain the question or comment differently, using expressions other than “dominant negative”. We would sincerely answer the question of the Reviewer and revise the manuscript accordingly.

4. Related to the point 3. As I understand it, the genetic library is constructed such that each transgenic gene is expressed from its original promoter. Essentially, this approach is designed to create an artificial CNVs by introducing an additional copy of a gene expressed episomally. Unfortunately such approach does not automatically guarantee expression of overexpression. Once again, this could lead to huge amounts of false negative results which render this approach not very impactful. It would be helpful to show how many genes/proteins were indeed overexpressed in the constructed libraries.

(Response 4)

We would like to thank the Reviewer for the comment. The centromere plasmid used in this study is maintained as a single copy DNA in the parasite via the unique function of the centromere. Therefore, *pfCRT* from Dd2 was maintained as a single copy gene in the selected transgenic parasites (Supplementary Fig. 4). In addition, *pfCRT* transcriptional levels were almost identical between 3D7 and Dd2 (Supplementary Fig. 5), which suggested that the *pfCRT* promoter activities from 3D7 and Dd2 were comparable in the selected parasites. Considering these results, we supposed that the expression levels of the genes including *pfCRT* on the delivered genomic DNA fragments were probably almost identical to those of the corresponding genes on the original genome. Therefore, we considered that the gene

expression levels of the delivered genomic DNA were at the physiological level and did not show overexpression.

In nature, parasites generally acquire drug resistance by increasing the copy number of the drug resistance genes from one to two via gene duplication. This fact suggests that the parasites need not overexpress the drug resistance genes for acquiring drug resistance and that approximately a 2-fold increase in the gene expression is sufficient. Therefore, we considered that the drug resistance gene that confers resistance by increasing its expression can be identified using our approach. Further, those conferring resistance by increasing not only the copy number but also the transcription level would be identified. In addition, the drug resistance genes that confer resistance via their mutation could be potentially identified. The mutations in the drug resistance genes often confer strong resistance to the parasite. For example, the resistance conferred by PfCRT and DHFR-TS mutations is approximately 20 and 100 times higher than that conferred by the corresponding genes, respectively, with respect to IC_{50} . Therefore, even if the expression of mutated drug resistance gene is at a physiological level, the transgenic parasite probably exhibits detectable drug resistance. Indeed, this study as well as our previous studies demonstrated that such genes could be identified via functional screening. In summary, we consider that even if the expression of genes on delivered genomic DNA fragments is at a physiological level, “gain-of-function-type” drug resistance genes could be identified using our approach.

In the revised manuscript, we have discussed that the expression of the genes on the delivered genomic DNA fragment is likely to be at a physiological level and also that this expression level is probably sufficient for functional screening (P.16. I. 359-369).

5. This issue is relevant for the subsequent part of the study in which *pfmdr7* gene was identified as a mefloquine resistance marker. In my view this result is somewhat misleading. Although the used the mefloquine resistant strain for the genetic library contraction, the obtained result does not explain its resistance mechanism. Instead, it shows that an artificially increased copy number of *pfmdr7* could lead to mefloquine resistance in any strain. Although that is interesting, the relevance of this observation for clinical setting is unclear at best. So far no clinical or epidemiological evidence was observed from *pfmdr7* involvement in drug resistance of *P. falciparum* and the authors did not try to find it. As such this could be a highly artificial observation of an in vitro phenotype. Also, this could be a lucky situation in which an increased copy number of *pfmdr7* gene drives resistance. This might not be the case for other possibly relevant markers that will be missed by this technique. Given how hard and labor intensive these experimental approach seems to be, this would not be my method of choice to search for genetic markers.

(RESPONSE 4)

In the present study, we collected the blood samples from patients living in the endemic area of the mefloquine-resistant parasite. The collection site, Mae Hong Son province, was the border area between

Thailand and Myanmar. These patients were treated using the artemisinin combination therapy (ACT), with piperazine as a partner drug. If we attempted to collect clinical data in this study, the patients would have been treated with mefloquine instead of ACT. However, this attempt might have endangered the patients, which could be never permitted due to the ethical considerations. This is the reason why we could not conduct clinical data collection. We examined the correlation between *pfmdr7* transcription and mefloquine resistance using the clinically-isolated parasites (Fig. 4b). The results showed *pfmdr7* upregulation in mefloquine-resistant but not in mefloquine-sensitive parasites. Although we could not conclude the clinical importance of *pfmdr7* upregulation in mefloquine resistance, this result suggested its possible involvement in mefloquine resistance in clinical samples.

Our additional experiment suggested that PfMDR7 could potentially confer mefloquine resistance and showed that an increase in the copy number of *pfmdr7* confers mefloquine resistance to the parasites (Fig. 4a). Therefore, the parasites would acquire mefloquine resistance if they increase their copy number or upregulate the expression. Another additional experiment identified 9 mutations in the *pfmdr7* promoter of the MEF1 strain compared with 3D7 and further showed that the *pfmdr7* promoter activity in MEF1 was 2.8–3.2-fold higher than that of *pfmdr7* in 3D7 (Fig. 4c). Similar upregulation was observed during the RT-qPCR analysis of *pfmdr7* of the transgenic parasites, in which *pfmdr7* genes from 3D7 and MEF1 were introduced together with their promoters (Supplementary Fig. S13). Therefore, these results strongly suggested that *pfmdr7* upregulation resulted from the mutations in the promoter. Considering these results, the resistant parasite possibly upregulates *pfmdr7* expression, which could naturally confer mefloquine resistance via mutations in its promoter, and pumps out mefloquine more efficiently compared with the sensitive parasite. We considered that this is the mechanism by which MEF1 acquires mefloquine resistance. In the revised manuscript, we described these results in the Results (P.12. l.268- P.13. l. 300) and Discussion sections (P.17. l.384- P.18.l.416).

Reviewer #2 (Remarks to the Author):

The approach described and validated in this manuscript constitutes a significant advance for the malaria field, providing a new means for the identification of drug resistance mechanisms in *Plasmodium falciparum*. While perhaps not novel in concept, given that the authors have validated the use of a similar approach in the genetically tractable murine malaria parasite *Plasmodium berghei* (Iwanaga et al. 2012 Genome Res), its adaptation for use with the most deadly human malaria parasite, *P. falciparum*, is important. For some antimalarial lead compounds, it is possible to generate resistant parasites readily in vitro in a matter of weeks and then sequence their genomes to determine the genetic basis for resistance. However, for other antimalarials and lead compounds, this is not possible. A low propensity for resistance development, and an inability to generate resistance readily in vitro, bodes well for the longevity of an antimalarial, and it is arguably such compounds that should be prioritized for clinical development. The

development in this study of a time-efficient approach to study resistance mechanisms for such compounds, that arise in the field or during the course of clinical trials, is extremely valuable.

The authors used their new approach to obtain interesting, novel data on mechanisms of mefloquine resistance. They identified *pfmdr7* as a resistance determinant for this clinically used drug. Interestingly, through well designed experiments, the authors provided evidence that transcriptional upregulation of *pfmdr7* (rather than the more common finding of copy number variation or SNPs) was the likely mechanism of mefloquine resistance in a clinical isolate. Such a resistance mechanism would not have been discovered with a whole genome sequencing approach, highlighting an additional major advantage of the authors' new approach to the identification of drug resistance genes.

The methodology is sound and well described. This reviewer only noted one potential weakness – that statistical analyses were not performed to compare the IC₅₀ values obtained in parasite proliferation assays for chloroquine or mefloquine between different parasite lines, possibly because independent biological replicates of these experiments were not performed. It is not clear whether the data from parasite proliferation assays shown in the Figures were derived from a single experiment or more. Ideally, multiple independent experiments should be performed for all parasite proliferation assays, such that IC₅₀ values can be provided as means with error values, and differences between parasite lines can be tested for statistical significance. It is noted however that the ability of parasites to survive in the presence of the drug provided complementary data enabling sensitive and resistant parasites to be distinguished from one another. At the minimum, the authors should state the number of biological and/or technical replicates from which the parasite proliferation data were obtained, in the relevant Figure legends.

On the whole, the manuscript is well written and the Figures are well presented. The methods are described in detail, which will allow similar experiments to be undertaken by other groups. Some minor suggestions for further improving clarity are outlined below:

To reviewer #2,

We greatly appreciate the careful revisions suggested by the Reviewer. We found the suggestions of the Reviewer extremely useful for the revision of our manuscript. Although we performed the assays for determining the IC₅₀ values in biologically independent quadruplicate experiments, we did not indicate the error values in the previous manuscript. In the revised manuscript, we included all error values for each IC₅₀ value (for example, P.8. I.168-I.171). In addition, we mentioned that those values were obtained from the quadruple experiments.

1. the concentration of pyrimethamine used to select for transfected parasites should be stated

Response1,

We added the final concentration of pyrimethamine that was used for the screening (P.5.I.105-I.107).

2. '> 10 kb of DNA fragments' or similar (used on two occasions) is not clear – consider replacing with 'DNA fragments > 10 kb in size'

Response2,

We would like to thank the Reviewer for the suggestion. We corrected the indicated formulation following the suggestion of the Reviewer (P.5.I.101, and P.15. I.338).

3. line 117: it is not clear whether the sizes referred to on this line include the vector size or not.

Response3,

The size of the signals detected using Southern analysis was the sum of the sizes of the insert DNA fragments and that of the vector. Therefore, for better clarity, we revised the sentence as follows:

"The signals in 10 parasite clones were detected at sizes in the range of 24.8–45.0 kb, including the size of the vector (8,018 bp; Supplementary Fig. S1c). The average size of the insert DNA fragments was estimated to be 25.9 kb after subtraction of the vector size."(P.7. I.136-I.141)

4. the phrase 'A clonal parasite was established from each surviving parasite' (used on two occasions) is not clear – consider replacing with 'A clonal parasite line was established from each culture containing surviving parasites'

Response4,

We amended them according to the suggestion of the Reviewer (P.8.I.165-I.166, and P.10. I.221-I.222).

5. lines 174-175: the data provided for the IC50s and fold difference appear to contradict one another

Response5,

We apologize for our careless mistake. We corrected it according to the suggestion of the Reviewer (P.10.I.212).

6. line 293: should this be 'efflux from parasites and/or sequestration into the digestive vacuole', given what is known about the localisation of PfMDR1?

Response6,

We completely agree with the comment of the Reviewer and have corrected the indicated text according to the suggestion of the Reviewer (P.18.I.401-I.402).

7. line 522: there is a word missing - L is the average [length?]

Response,

We apologize for our careless mistake. We corrected the indicated part (P.29.l.660).

8. Supp. Fig. 7: is Day 0 when the drug was first added? It would be good to clarify.

Response,

We would like to thank the Reviewer for the comment. Day 0 is the first day for the drug treatment period. This graph referred to the parasite survival curve used for the IC₅₀ value and the concentration used for the assay was indicated as log₁₀(concentration). Therefore, "0" was not indicated in Supplementary Figure S14 (previous Supplementary Figure S7).

Reviewer #3 (Remarks to the Author):

This manuscript describes a potentially powerful approach to identifying the gene(s) underlying drug resistance in the human malaria parasite. The system includes generating a resistant parasite genomic library with large (>10 kb) inserts in an artificial chromosome vector. The library will be directly transfected to a sensitive lab parasite strain (in their case, 3D7). The authors have shown that they could obtain an average of ~500 transfectants with an average insert size of 25 kb, roughly covering half of the genome. Two repeated transfections would cover one genome equivalence. These library of transfectants are subjected to selection with the drug of interest, and emerging clones can be genotyped to determine the insert genes that might be responsible for the resistance phenotype. As a proof of principle, they have demonstrated that this approach works for determining pfcr as the determinant of chloroquine resistance. Then they used a field parasite isolate from western Thailand showing low-level resistance to mefloquine (MQ) to identify that *mdr7* was responsible for the resistance to MQ.

To reviewer #3,

We greatly appreciate the important comments and meaningful suggestions provided by the Reviewer. These suggestions and questions were useful for designing our supplementary experiments and revising our manuscript. Please find our point-by-point responses below.

Major comment:

I agree with the authors that this will be a very useful functional method to identify drug resistance genes. However, I also feel that the MQ resistance part is a bit weak, and deserves improvement. The identification of 4 clones with overlapping chromosome 12 fragments is quite convincing, but the subsequent experiments did not substantially corroborate the data that *mdr7* is the responsible gene. 1)

It did not show the reason why *mdr7* had higher expression in the resistant strains? They did not identify mutations in the ORF, but what about the promoter region? For the sequence introduced into 3D7, does it contain the promoter sequence? If yes, is it different from 3D7? 2) For the *mdr7* gene introduced into 3D7, is the extra copy from the artificial chromosome responsible for the overexpression? If so, I would assume that inserting an extra copy of the 3D7 *mdr7* gene might have the same effect. 3) The 1.6x increase in *mdr7* gene expression is weak evidence supporting the conclusion. I wonder whether the authors could express *mdr7* ORF using different promoters (with different strengths) in their artificial chromosome to further demonstrate this?

Response 1

Nine mutations could be identified in the *pfmdr7* promoter region of MEF1 (Supplementary Fig. S11). We used this mutated promoter when we generated transgenic parasites, and the *pfmdr7* of MEF1 was introduced into these parasites. We further generated transgenic parasites in which the *pfmdr7* of 3D7 was introduced along with its promoter in a supplementary experiment. The RT-qPCR analysis of *pfmdr7* in those two transgenic parasites showed that the transcription of the *pfmdr7* gene in MEF1 was approximately 3.5-fold higher than that in 3D7 (Supplementary Fig. S13). Furthermore, we compared the *pfmdr7* promoter activities between MEF1 and 3D7 via reporter assay using two luciferases. The results showed that the *pfmdr7* promoter activity in MEF1 was 2.8–3.2 fold higher than that in 3D7 (Fig. 4c). Therefore, these results suggest that mutations in the *pfmdr7* promoter participated in its upregulation in MEF1. We considered that these mutations probably conferred mefloquine resistance to MEF1 (P.13.I.285-I.300).

Response 2

We would like to thank the Reviewer for the important comment. To address the question, we introduced the CDS of PfMDR7 in 3D7 along with its promoter into 3D7 as described in response 1. The transgenic parasite acquired mefloquine resistance via the introduction of an extra copy of the *pfmdr7* gene, as pointed out by the Reviewer (Fig. 4a). These results showed that PfMDR7 has the potential to reduce the pharmacological activity of mefloquine (P.12. I.274- P.13.I.282).

Response 3

We had a similar idea as that of the Reviewer and attempted to express *pfmdr7* gene using strong and moderate promoters, originating from the *pfhsp70* (PF3D7_0818900) and *calmodulin* (PF3D7_1434200) genes, respectively. Briefly, each promoter was cloned upstream of the CDS of *pfmdr7* on the centromere plasmid, and the strain 3D7 was transfected with those plasmids. This centromere plasmid exhibits the human dihydrofolate-reductase gene as a drug selectable marker, and the transgenic parasites harboring those plasmids were thus screened using pyrimethamine. A similar plasmid was used for generating the

transgenic parasite in which an extra copy of *pfmdr7* was introduced along with the original promoter. In contrast to the case using the *pfmdr7* promoter, when the promoters of *pfhsp70* and *calmodulin* were used, all parasites died completely during pyrimethamine-screening in two independent experiments. These results strongly suggested that *pfmdr7* overexpression was lethal for the parasites. Therefore, appropriate *pfmdr7* upregulation is probably essential for conferring mefloquine resistance to the parasites. In contrast, significant upregulation might impose a huge fitness cost to the parasites.

Minor comments:

1. In the second paragraph, the authors stated the lengthy process of identifying k13 as the gene mediating artemisinin resistance and the desire for an alternative approach to speed up this process. I think this point needs to be discussed, since the functional screening approach described here may not be suitable for identifying artemisinin resistance genes given the constant drug pressure needed to screen the transfectant library.

Response

We would like to thank the Reviewer for the comment. We agree with the opinion of the Reviewer. In the previous manuscript, we used the case of artemisinin-resistant parasites as an example to explain that producing resistant parasites is difficult *in vitro*. However, we believe that this sentence could potentially confuse the readers. Therefore, we deleted this sentence from the revised manuscript and included the following sentence:

“However, it generally takes months to several years to obtain drug-resistant parasites using drug exposure techniques, which is one of reasons why the identification of drug resistance genes requires a long time.” (P.3.I.60-I.63)

2. Line 100: The manuscript style presents results first. It would make it easier to understand what was used as the target parasite without going to the method section in the back (in this case, *P. falciparum* 3D7).

Response

We indicated the name of the parasite strains in the Results section according to the suggestion of the Reviewer (for example, P5. I.100).

3. Line 102: explain why pyrimethamine selection (dhfr cassette in the artificial chromosome vector?)

Response

As the Reviewer commented, the centromere plasmid contains the human dhfr cassette, allowing for the pyrimethamine selection of the transgenic parasites. For better clarity, we described in the Results section that pFCENV1 contains the human dhfr gene as a pyrimethamine selectable marker (P.5.I.105- I.107).

4. Paragraph starting at line 210: it is difficult to understand how this was done. Which part of each gene was cloned and expressed? Does that include the promoter and 3' region? This information is critical for the readers to understand how this would increase *mdr7* expression?

Response

We cloned the CDS and promoters of each identified gene (PF3D7_1209800 [*atp11*], PF3D7_1209900 [*pfmdr7*], and PF3D7_1210000 [*rp11*]) into the centromere plasmid. The *pbhsp70* 3'UTR was used for the transcriptional termination of those genes. Although it originated from the rodent malaria parasite, *P. berghei*, it functions in *P. falciparum*. The resulting plasmids were introduced into the strain 3D7, and the transgenic parasites harboring each plasmid were then screened using pyrimethamine. We included the above information in the Results section of the revised manuscript (P.11. I.248-I.267).

5. Some comments on validation using this approach would be nice. Since the putative gene might be large (for example, *mdr7* ORF plus promoter/3 region may exceed 5-6 kb), amplifying the large piece and inserting it in the vector might not be easy. Thus, this caveat may be mentioned, given that many plasmodium genes are large.

Response

As the Reviewer commented, the PF3D7_1209800 and PF3D7_1210000 genes were too large to be incorporated into the centromere plasmid using *E. coli*. Therefore, we ligated the DNA fragments of those genes with the digested centromere plasmid *in vitro*, and those ligation reaction mixtures were then directly introduced into 3D7. We described this transgenic parasite generation in the Materials and Methods section (P.36.I.833-I.844). Further, we included this information in Results section of the revised manuscript (P.13. I.258-I.262,).

REVIEWERS' COMMENTS

Reviewer #1 (Remarks to the Author):

All my concerns were addressed. Given the positive opinions of the remaining reviewers, I agree with publishing this manuscript in the current form.

Reviewer #2 (Remarks to the Author):

The manuscript has improved since the original submission, and represents an important advance. The evidence that *pfmdr7* expression level may be involved in mefloquine resistance has been strengthened, and the authors are appropriately cautious in their interpretation.

One question that the authors might like to consider commenting on in the Discussion - why was *pfmdr1* not confirmed as a mefloquine resistance gene in their experiments? Would the introduction of a second copy of *pfmdr1* be expected to generate a high enough level of resistance? Perhaps it depends on which isoforms are expressed by 3D7 and MEF1?

Reviewer #3 (Remarks to the Author):

The authors have done additional experiments to strengthen their claim and addressed all my comments.

Editorial revision may be needed to smooth the English.